# Suppression of SRCAP chromatin remodelling complex and restriction of lymphoid lineage commitment by Pcid2

Buqing Ye[1], Benyu Liu[1,2], Liuliu Yang[1,2], Guanling Huang[1,2], Lu Hao[1,2], Pengyan Xia[1], Shuo Wang[1], Ying Du[1], Xiwen Qin[1,2], Pingping Zhu[1], Jiayi Wu[1,2], Nobuo Sakaguchi[3], Junyan Zhang[4] & Zusen Fan[1,2]

Lymphoid lineage commitment is an important process in haematopoiesis, which forms the immune system to protect the host from pathogen invasion. However, how multipotent progenitors (MPP) switch into common lymphoid progenitors (CLP) or common myeloid progenitors (CMP) during this process remains elusive. Here we show that PCI domain-containing protein 2 (Pcid2) is highly expressed in MPPs. *Pcid2* deletion in the haematopoietic system causes skewed lymphoid lineage specification. In MPPs, Pcid2 interacts with the Zinc finger HIT-type containing 1 (ZNHIT1) to block Snf2-related CREBBP activator protein (SRCAP) activity and prevents the deposition of histone variant H2A.Z and transcription factor PU.1 to key lymphoid fate regulator genes. Furthermore, *Znhit1* deletion also abrogates H2A/H2A.Z exchange in MPPs. Thus Pcid2 controls lymphoid lineage commitment through the regulation of SRCAP remodelling activity.

[1] Key Laboratory of Infection and Immunity of CAS, CAS Center for Excellence in Biomacromolecules, Institute of Biophysics, Chinese Academy of Sciences, Beijing 100101, China. [2] University of Chinese Academy of Sciences, Beijing 100049, China. [3] WPI Immunology Frontier Research Center, Osaka University, Osaka 565-0871, Japan. [4] Harvard Stem Cell Institute, Harvard Medical School, Boston, MA 02115, USA. Buqing Ye and Benyu Liu contributed equally to this work. Correspondence and requests for materials should be addressed to Z.F. (email: fanz@moon.ibp.ac.cn)

Adult haematopoiesis depends on a rare population of haematopoietic stem cells (HSC) in the bone marrow (BM) that possess the capacity for self-renewal and differentiation[1]. HSCs comprise long-term HSCs (LT-HSC) and short-term HSCs (ST-HSC). LT-HSCs, at the very top of the cellular hierarchy, are endowed with the ability to continuous supply of blood cells owing to their self-renewal and differentiation[2,3]. ST-HSCs, losing self-renewal ability, are doomed to differentiate and give rise to multiple blood cell lineages. Multipotent progenitors (MPPs), a downstream progenitor of

ST-HSCs, can generate either common lymphoid progenitors (CLPs) or common myeloid progenitors (CMPs)[4–6]. CLPs produce all lymphoid cells but lose myeloid potential[7], whereas CMPs give rise to myeloid cells and lose lymphoid capacity[8]. The differentiation into lymphoid- or myeloid-restricted progenitors are tightly controlled by intrinsic and extrinsic signals[9,10]. However, the underlying mechanism regulating MPP fate decisions into CLPs or CMPs remains elusive.

Pcid2 (PCI-domain containing protein 2) is a homologue of yeast protein Thp1 that participates in the export of mRNAs from the nucleus to cytoplasm[11]. A report showed that Pcid2 is in the human TREX2 complex and prevents RNA-mediated genome instability[12]. Through genome-scale RNA interference (RNAi) screening, Pcid2 was identified to be an important factor that is involved in the self-renewal of mouse embryonic stem cells (ESCs)[13]. We demonstrated that Pcid2 modulates the pluripotency of mouse and human ESCs via regulation of EID1 protein stability[14]. Moreover, Pcid2 is selectively involved in the transport of MAD2 mRNA that modulates the mitotic checkpoint during B-cell development[15]. However, how Pcid2 modulates the HSC fate decision in mammalian haematopoiesis is still unclear.

During differentiation, the haematopoietic lineage development follows a strict hierarchical pattern programming emanating from a few HSCs. Both genetic and epigenetic modulations are involved in the regulation of haematopoietic lineage specification[16,17]. DNA organized in loose chromatin (euchromatin) is readily available for gene expression, while DNA tightly packed into dense chromatin (heterochromatin) becomes inaccessible to genetic reading and transcription. Chromatin remodelling is a prerequisite for eukaryotic gene transcription[18], which relies on ATP-dependent remodelling complexes. These remodelling complexes are divided into four major subfamilies, including SWI/SNF, ISWI, CHD and INO80 subfamilies, based on a common SWI2/SNF2-related catalytic ATPase subunit[19,20]. The SNF2-related CBP activator protein (SRCAP)-contained remodelling complex, termed SRCAP complex, belongs to the INO80 subfamily. Eleven protein subunits, including SRCAP, ZNHIT1, Arp6, and YL-1, have been identified in the SRCAP complex[21]. The SRCAP complex can exchange histone H2A for the variant H2A.Z in the nucleosomes, rending accessible DNA for gene transcription[22]. H2A.Z is proposed to activate target gene transcription enhancing the promoters' accessibility of the target genes[23]. Moreover, in the haematopoietic system, increased H2A.Z serves as a chromatin signature during the differentiation of haematopoietic stem or progenitor cells[24].

Here we show that Pcid2 is highly expressed in the BM and restricts lymphoid lineage specification. PCID2 binds to ZNHIT1 to block the SRCAP complex remodelling activity and prevents H2A.Z/H2A exchange of key lymphoid fate regulator genes in MPPs, leading to skewed lymphoid lineage commitment.

## Results

### Pcid2 knockout (KO) increases lymphoid but decreases myeloid cells.

We reported that Pcid2 inactivates developmental genes to sustain the pluripotency of mouse and human ESCs via regulation of EID1 stability[14]. We next sought to explore whether Pcid2 is involved in the haematopoiesis. We noticed that Pcid2 was most highly expressed in BM and haematopoietic progenitor cells, whereas it was almost undetectable in mature blood cells (Fig. 1a, and Supplementary Fig. 1A), suggesting that Pcid2 may have a function in the regulation of haematopoiesis. Since Pcid2 KO causes early embryonic lethality[14], we thus crossed Pcid2[flox/flox] mice with MxCre mice to generate Pcid2[flox/flox];MxCre[+] mice. Cre recombinase expression was induced by poly (I:C) treatment for three times. Pcid2 was completely deleted in BM after poly (I:C) treatment (Fig. 1b; Hereafter, poly (I:C)-treated Pcid2[flox/flox];MxCre[+] mice are called as Pcid2[−/−], whereas poly(I:C)-treated Pcid2[+/+];MxCre[+] mice are referred to as Pcid2[+/+]). We observed that Pcid2[−/−] mice displayed decreased BM cellularity by 8 week deletion of Pcid2 (Fig. 1c). Of note, Pcid2[−/−] mice had an enlarged thymus and cervical lymph nodes (Fig. 1d), as well as increased numbers of lymphocytes (Fig. 1e). Consistently, we observed that Pcid2[−/−] mice displayed significantly (P < 0.01, t-test) increased thymic and cervical lymph node CD4[+] and CD8[+] T cells compared with Pcid2[+/+] mice (Fig. 1e and Supplementary Fig. 1B). In addition, Pcid2[−/−] mice had an increased number of lymphocytes in peripheral blood compared with Pcid2[+/+] littermate control mice (Fig. 1f and Table 1). Furthermore, Pcid2[−/−] mice had increased numbers of CD3[+] T cells and NK1.1[+] natural killer (NK) cells in peripheral blood compared with littermate wild-type (WT) control mice (Fig. 1f and Supplementary Fig. 1C). Notably, Pcid2[−/−] mice displayed an about 2-fold decreased numbers of granulocytes and monocytes compared with littermate control mice (Fig. 1f and Table 1). By contrast, Pcid2[−/−] mice had similar counts of red blood cells and platelets comparable to littermate WT control mice (Fig. 1f and Table 1). To exclude the effect of interferon (IFN), we also crossed Pcid2[flox/flox] mice with Vav-Cre mice (an IFN-independent Cre system) to generate Pcid2[flox/flox];Vav-Cre[+] mice. We observed that Pcid2[flox/flox];Vav-Cre[+] mice displayed the same phenotype as Pcid2[flox/flox];MxCre[+] mice after poly (I:C) treatment (Supplementary Fig. 1d). These data suggest that Pcid2 deficiency causes skewed lymphoid cell differentiation.

Given that blood mature cells are differentiated from BM progenitors, we then tested the changes of BM progenitors. We found that Pcid2[−/−] mice exhibited decreased LSK (Lin[−]Sca-1[+]c-

**Fig. 1** Pcid2 KO increases lymphoid cells but decreases myeloid cells. **a** Total RNA was extracted from the indicated tissues and analysed by real-time qPCR. Primer pairs are shown in Supplementary Table 1. BM bone marrow. **b** Conditional Pcid2 KO mice were generated as described in Methods section. **c** Paraffin sections from the femurs of 6-week-old mice were stained with haematoxylin and eosin (H&E) staining. Scale bars, 20 μm. BM cellularity per femur was enumerated by flow cytometric single cell counting and shown as means ± S.D. n = 6 for each group. **d** Representative photographs of thymus (upper panel) and cervical lymph node (lower panel) were obtained. Relative weight of organs was identified as a percentage of total body weight and shown as means ± S.D. (n = 5). **e** Thymus and cervical lymph node sections were stained by H&E. Scale bar, 50 μm. Cell numbers and percentages of T cells were analysed by flow cytometry (n = 5). **f** Lymphocytes (Lym), granulocytes (Gran), monocytes (Mon), red blood cells (RBC) and platelets (PLT) were analysed by XFA6030 automated haemacytometer (Slpoo). Results are shown as means ± S.D. n = 6., Percentages of CD3[+] T cells and NK1.1[+] NK cells in PBMCs were analysed by flow cytometry and shown as means ± S.D. (n = 5). **g** Flow cytometric analysis of Lin[−]Sca1[+]cKit[+] (LSK,), multipotent progenitor (MPP), long-term HSC (LT-HSC), lymphoid-primed multipotent progenitor (LMPP), common lymphoid progenitor (CLP) and common myeloid progenitor (CMP) from BM. Number of indicated cells was calculated and shown as means ± S.D. (n = 5). **h** Flow cytometric analysis of pro T subtypes of DN1, DN2, DN3 and DN4 from mouse thymus and calculated as in **g**. (n = 5). **i** Flow cytometric analysis of pre-pro B, pro B and NK cell progenitors (NKP) from BM and calculated as in **g**. (n = 5). **j** Flow cytometric analysis of granulocyte-macrophage progenitor (GMP) and megakaryocyte–erythrocyte progenitor (MEP from BM and calculated as in **g**, (n = 5). **k** Quantitative image analysis of the MPP and CLP in femurs. Cell counts were enumerated in right panel graph (n = 20 fields). Scale bar, 500 μm. Red arrowheads denote MPPs; yellow arrowheads denote CLPs. Results are shown as means ± S.D. **P < 0.01. NS, no significance. Student's t-test was used as statistical analysis. Data are representative of at least three independent experiments

**Table 1 Haematopoietic parameters**

| Parameter | MxCre+; Pcid2+/+ (Ctrl), n = 10 | MxCre+; Pcid2flox/flox (KO), n = 10 |
|---|---|---|
| RBC (x10^6/μl) | 9.3 ± 0.5 | 9.0 ± 0.6 |
| Platelet (x10^3/μl) | 913.2 ± 98.7 | 899.6 ± 8.7 |
| Lymphocytes (×10^3/μl) | 6.1 ± 0.1 | 9.2 ± 0.1 |
| Granulocytes (×10^3/μl) | 5.2 ± 0.3 | 2.1 ± 0.1 |
| Monocytes (x10^3/μl) | 1.1 ± 0.02 | 0.5 ± 0.01 |

*PLT* platelet, *RBC* red blood cell
20 μl blood per sample was analysed using an XFA6030 automated haemacytometer (Slpoo). Cell numbers and percentages were counted for each population. Differential WBCs were determined by monitoring the morphology of cells stained with May-Gruenwald Giemsa. Data are shown as means ± S.D

Kit+) and MPP (Lin−Sca-1+c-Kit+CD150−CD48+) cell counts (Fig. 1g and Supplementary Fig. 2A). By contrast, LT-HSC and lymphoid-primed multipotent progenitor (LMPP, Lin−Sca-1+c-Kit+CD150−Flt3+) were comparable in $Pcid2^{-/-}$ mice and $Pcid2^{+/+}$ WT mice. As expected, Pcid2 deficiency caused an about 3-fold increased number of CLP (Lin−IL7Rα+c-Kit^low Sca-1^low Flt3+) cells compared with WT control mice, whereas an around 2-fold decreased counts of CMP (Lin−c-Kit+Sca-1−CD34+CD16/32−) cells in $Pcid2^{-/-}$ mice. Consistently, $Pcid2^{flox/flox}$;Vav-Cre+ mice displayed the same phenotype as $Pcid2^{flox/flox}$;MxCre+ mice after poly (I:C) treatment (Supplementary Fig. 2B). In addition, we analysed MPP populations using CD229 and CD244 along with SLAM markers. We found that numbers of MPP-1 (CD150−CD48−CD229−CD244−LSK), MPP-2 (CD150−CD48−CD229+CD244− LSK), and MPP-3 (CD150−CD48−CD229+CD244+LSK) were all decreased in Pcid2 KO mice compared to $Pcid2^{+/+}$ mice (Supplementary Fig. 2B).

We then analysed double negative (DN) 1–4 compartment cells in $Pcid2^{+/+}$ and $Pcid2^{-/-}$ thymocytes. DN subpopulations are classified as DN1 (CD44+CD25−), DN2 (CD44+CD25+), DN3 (CD44−CD25+) and DN3 (CD44−CD25−). We noticed that numbers of DN1–4 compartment cells in $Pcid2^{-/-}$ thymocytes were all increased compared to $Pcid2^{+/+}$ thymocytes (Fig. 1h and Supplementary Fig. 2C). Moreover, pre-pro B (B220+CD43+ BP-1−HSA−), pro B (B220+CD43+BP-1−HAS+) and NK cell progenitors (NKP, Lin−CD122+DX5−NK1.1−) were all increased in $Pcid2^{-/-}$ mice (Fig. 1i and Supplementary Fig. 2C). By contrast, numbers of megakaryocyte–erythrocyte progenitor (MEP, Lin−c-Kit+Sca-1−CD34−CD16/32−) in $Pcid2^{-/-}$ mice were decreased by 1.4-fold and numbers of granulocyte-macrophage progenitor (GMP, Lin−c-Kit+Sca-1−CD34+CD16/32+) in $Pcid2^{-/-}$ mice were decreased by 1.5-fold in BM (Fig. 1j and Supplementary Fig. 2D). These observations were further validated in BM by immuno-fluorescence staining (Fig. 1k). Altogether, Pcid2 deficiency increases lymphoid cells but decreases myeloid cells.

**Enhanced lymphoid differentiation of Pcid2-deficient MPPs.** We showed above that Pcid2 deficiency caused increased CLPs and decreased CMPs. Their upstream progenitor MPPs were also markedly declined in Pcid2-deficient mice. Ki67 staining can discriminate cells in the G0 phase of the cell cycle from those in the cycling state[25]. We noticed that MPPs exhibited no obvious alteration of resting state (G0) vs. cycling state (S/G2/M) ratio between Pcid2 KO and WT littermate control mice (Fig. 2a). In addition, MPPs, CMPs and CLPs did not undergo apparent apoptosis in Pcid2 KO vs. WT control mice (Fig. 2b). These results suggest that the skewed ratio of the CLPs and CMPs in Pcid2 KO mice may be caused by lineage specification of the MPP progenitor.

We next sorted LMPPs from BM of $Pcid2^{-/-}$ mice and littermate WT control mice and co-cultured them with OP9-DL1

stromal cells[26]. We observed that $Pcid2^{-/-}$ LMPPs produced all increased DN1–4 compartment cells (Fig. 2c). Of note, the ratio of each DN subset between $Pcid2^{-/-}$ and $Pcid2^{+/+}$ LMPPs was not changed, suggesting that Pcid2 KO does not affect the stages of DN1–4s. With reduced interleukin (IL)-7 concentrations from culture media, $Pcid2^{-/-}$ LMPPs differentiated into more mature T cells (Fig. 2d). These results indicate that $Pcid2^{-/-}$ LMPPs prefer to differentiate into lymphoid cells. We also sorted $Pcid2^{+/+}$ and $Pcid2^{-/-}$ MPPs and co-cultured them with OP9 stromal cells to induce B-cell and myeloid cell generation. We observed that $Pcid2^{-/-}$ MPPs generated reduced B cells and myeloid cells compared with $Pcid2^{+/+}$ MPPs (Fig. 2e), which were consistent with the report by Nakaya and colleagues[26]. These results suggest that Pcid2 deficiency suppresses the differentiation of mature B cells and myeloid cells but enhances the specification of mature T cells. We then conducted haematopoietic colony-forming cell (CFC) assays to test myeloid lineage colony formation. We noticed that Pcid2-deficient HSCs formed much fewer myeloid lineage colonies in vitro, including CFU-GEMM (colony-forming unit-granulocyte, erythroid, macrophage, megakaryocyte), CFU-M (colony-forming unit-macrophage), CFU-G (colony-forming unit-granulocyte), CFU-GM (colony-forming unit-granulocyte macrophage) and BFU-E (burst-forming unit-erythroid) colonies, than $Pcid2^{+/+}$ HSCs (Fig. 2f).

To further confirm that Pcid2 deficiency induced lymphoid lineage commitment in vivo, we sorted LT-HSCs from the BM of $Pcid2^{-/-}$ mice and transplanted them into lethally irradiated recipient mice (CD45.1). Donor-derived BM haematopoietic progenitor cells in recipient mice were analysed 8 weeks after transplantation. We rescued Pcid2 in $Pcid2^{-/-}$ LT-HSCs via infection with Pcid2-overexpressing retrovirus. Pcid2 was restored in $Pcid2^{-/-}$ LT-HSCs, which was comparable to that of WT LT-HSCs (Fig. 2g). We found that Pcid2-deficient HSC transplantation caused declined BM cellularity (Fig. 2h), as well as decreased numbers of MPPs and CMPs, but increased numbers of CLPs in the BM (Fig. 2i and Supplementary Fig. 3A). Additionally, Pcid2-deficient HSC transplantation resulted in increased numbers of CD3+ T cells and NK1.1+ NK cells but decreased numbers of granulocytes and monocytes in peripheral blood (Fig. 2j and Supplementary Fig. 3B). We observed that transplantation of Pcid2-rescued LT-HSCs could restore the biased lymphoid differentiation caused by Pcid2 deletion (Fig. 2h–j). Collectively, Pcid2 is involved in the regulation of branchpoint lineage commitment of MPPs.

**Cell-intrinsic functions of Pcid2 in lymphoid commitment.** To further verify whether Pcid2 deficiency-mediated biased lymphoid lineage differentiation is intrinsic or extrinsic, we transplanted BM cells from donor mice ($Pcid2^{flox/flox}$;MxCre+ or $Pcid2^{+/+}$; MxCre+ mice, CD45.2) into lethally irradiated recipient mice (CD45.1) (Fig. 3a). After successfully reconstituting the recipient BM, poly(I:C) was administered to induce deletion of the Pcid2 gene, followed by examination of lymphoid and myeloid cells. We found that Pcid2-deficient BM transplantation into recipient mice resulted in reduced numbers of LSKs, MPPs and CMPs but increased numbers of CLPs in the BM (Fig. 3b). Moreover, Pcid2-deficient BM engraftment also caused increased numbers of CD3+ T cells and NK1.1+ NK cells but decreased counts of CD11b+Gr1+ granulocytes (Fig. 3c). We next transplanted BM cells from WT donor mice (WT, CD45.1) into lethally irradiated recipient mice ($Pcid2^{flox/flox}$;MxCre+ or $Pcid2^{+/+}$;MxCre+ mice, CD45.2) for reciprocal transplantation (Supplementary Fig. 4A). We noticed that WT CD45.1 BM could successfully reconstitute BM of both $Pcid2^{flox/flox}$;MxCre+ and $Pcid2^{+/+}$;MxCre+ mice, and $Pcid2^{flox/flox}$;MxCre+ reconstituted BM exhibited a comparable

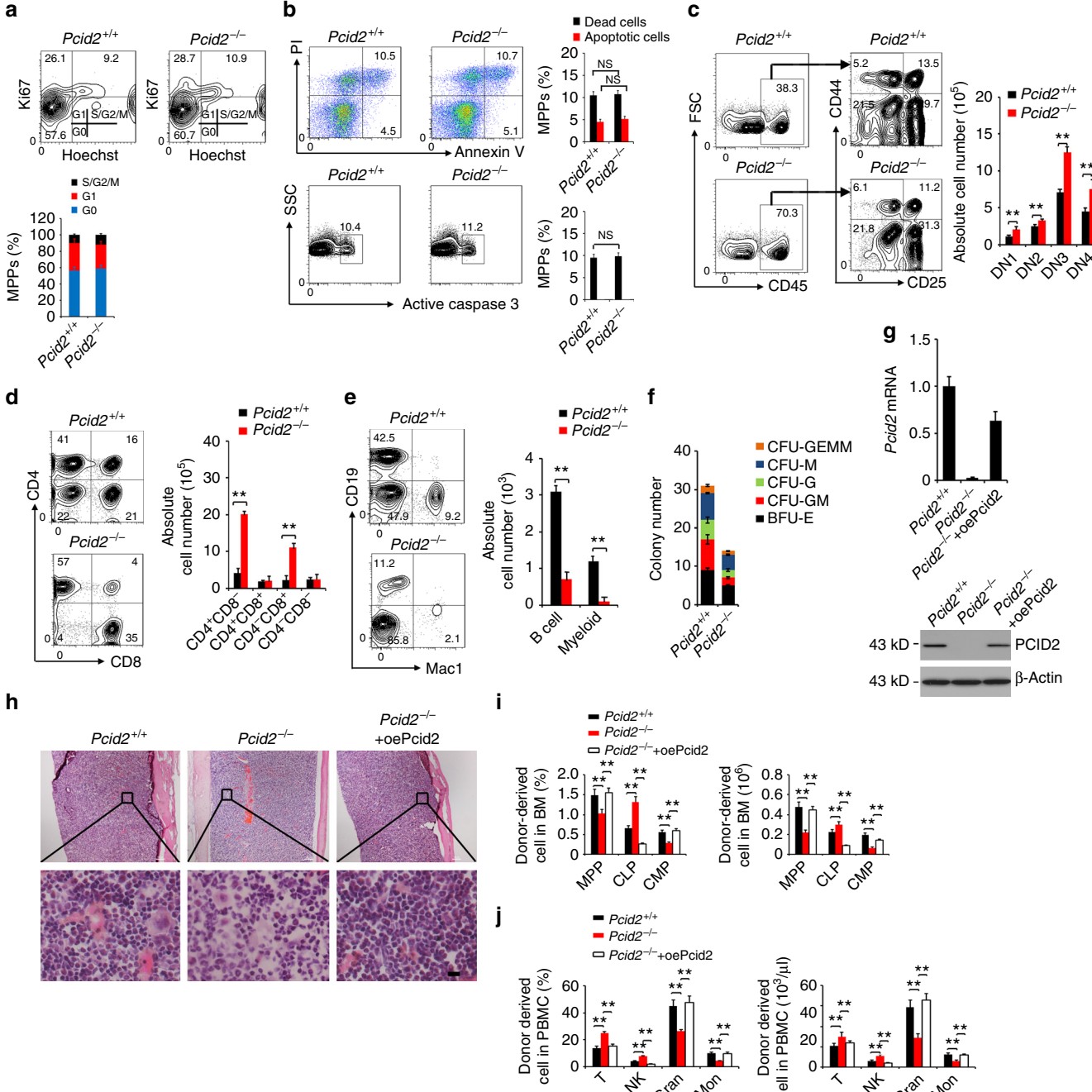

**Fig. 2** *Pcid2*-deficient MPPs prefer to differentiate into lymphoid cells but not myeloid cells. **a** Cell cycle analysis of MPPs from *Pcid2*[+/+] and *Pcid2*[−/−] mice. Cells were stained by Ki67 and Hoechst 33342 followed by flow cytometry. Percentages of cell cycle distributions were calculated and shown as means ± S.D. **b** MPPs from *Pcid2*[+/+] and *Pcid2*[−/−] mice were stained with Annexin V/PI or detected by active caspase 3. NS no significance. **c** Lymphoid-primed multipotent progenitors (LMPP) were sorted and co-cultured with OP9-DL1 stromal cells in the presence of Flt3-ligand and IL-7. After 14 days, cells were harvested and analysed by flow cytometry. **d** Cells were sorted and cultured as in **c** in the presence of Flt3-ligand and IL-7 and used Flt3 ligand alone thereafter. After 4 days, cells were harvested and analysed by flow cytometry. CD45-positive indicated cells were counted as absolute cell number and were shown as means ± S.D. **P < 0.01. **e** Multipotent progenitors (MPP) from BM of *Pcid2*[+/+] or *Pcid2*[−/−] mice were co-cultured with OP9 stromal cells in the presence of Flt3-ligand, IL-7 and SCF, followed by flow cytometry. CD45-positive cells were counted as absolute cell numbers and were shown as means±SD. **P < 0.01. **f** MPPs from BM of *Pcid2*[+/+] or *Pcid2*[−/−] mice were cultured in methocult medium for colony-forming cell (CFC) formation assays. CFU-GEMM, CFU-M, CFU-G, CFU-GM and BFU-E colonies were counted, respectively. Results are shown as means ± S.D. n = 5 for each group. **g** 1 × 10³ HSCs were isolated from BM of *Pcid2*[+/+] or *Pcid2*[−/−] mice and infected with pMY-GFP-Pcid2 (oePcid2) or empty vector containing retrovirus and transplanted into lethally irradiated recipient mice together with 5 × 10⁶ helper cells. Pcid2 restoration was confirmed by quantatitive PCR and immunoblotting. **h** Paraffin sections from the femurs were stained with H&E 8 weeks after transplantation. Scale bars, 20 μm. **i** BM progenitor populations from donor mice (GFP⁺) were analysed 8 weeks after transplantation. n = 6 for each group. **P < 0.01. **j** Peripheral blood cell populations from donor mice were analysed 8 weeks after transplantation. n = 6 for each group. **P < 0.01. Student's t-test was used as statistical analysis. Data are representative of at least three independent experiments

blood constitute to $Pcid2^{+/+}$;MxCre$^+$ reconstituted BM (Fig. 3d and Supplementary Fig. 4B, C). We next carried out competitive transplantation experiments. 1:1 mixture of $5 \times 10^2$ donor

$Pcid2^{-/-}$ LT-HSCs (CD45.2) and $5 \times 10^2$ donor $Pcid2^{+/+}$ LT-HSCs (CD45.1/CD45.2) together with $5 \times 10^6$ helper BM cells (CD45.1/CD45.2) were transplanted into lethally irradiated

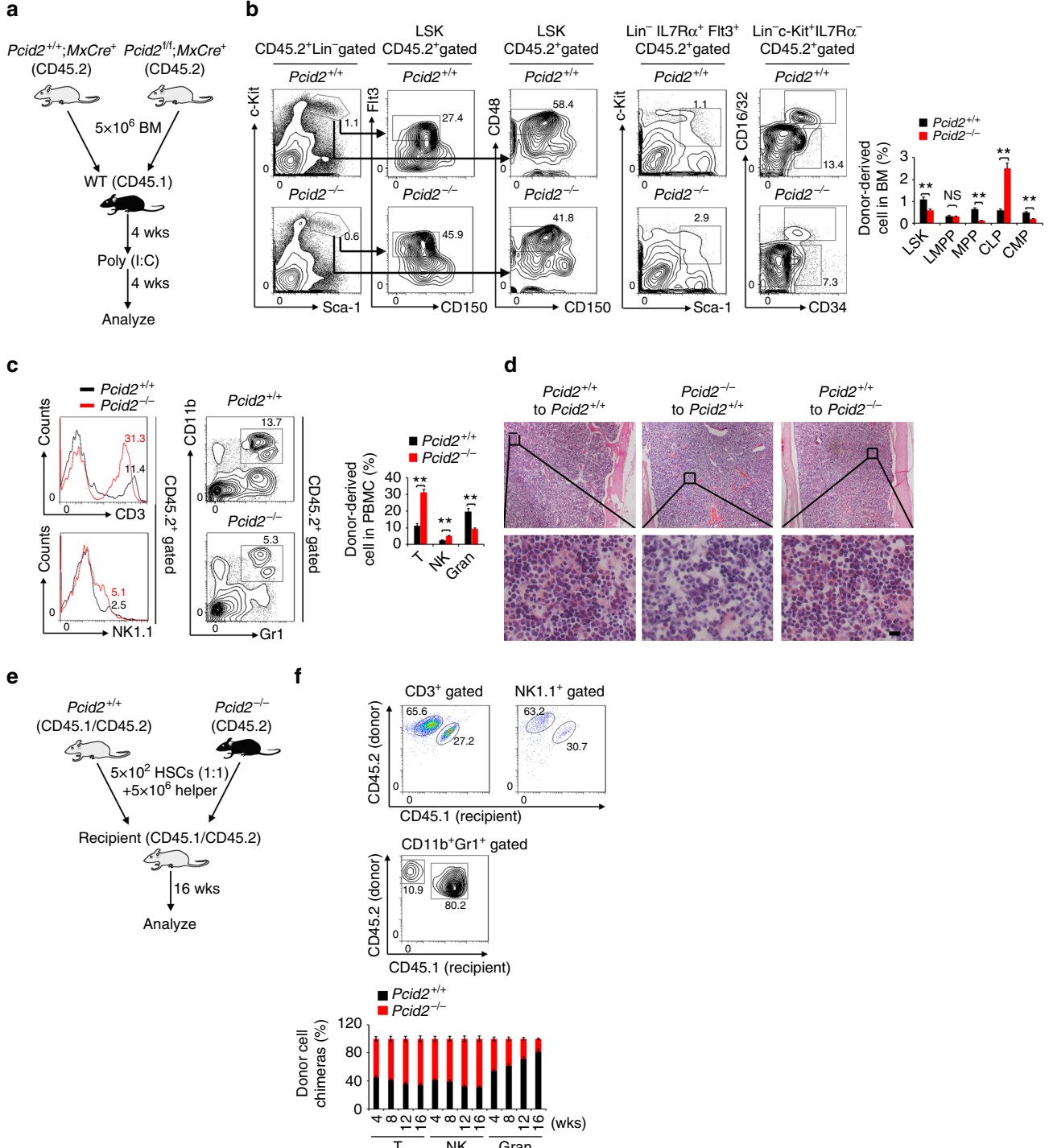

**Fig. 3** $Pcid2$ deficiency-mediated bias of lymphoid lineage commitment is cell intrinsic. **a** Schematic representation of transplantation. BM cell populations of recipient mice were analysed 8 weeks after transplantation. **b** Flow cytometric analysis of LSKs, MPPs, LMPPs, CLPs and CMPs from the BM of donor mice. Percentages of the indicated progenitor cells are shown as means ± S.D (right panel). $n = 6$ for each group. **$P < 0.01$. **c** Flow cytometric analysis of T cells, NK cells and granulocytes from the peripheral blood of donor mice. $n = 6$ for each group. **$P < 0.01$. **d** Representative paraffin sections from the femurs of the indicated recipient mice were stained with H&E. Scale bars, 20 μm. **e** Schematic representation of competitive BM transplantation. 1:1 mixture of CD45.1$^+$CD45.2$^+$ wild-type and CD45.2$^+$ $Pcid2^{-/-}$ HSCs was transplanted into lethally irradiated CD45.1$^+$/CD45.2$^+$ recipients together with $5 \times 10^6$ helper BM cells. BM cell populations of recipient mice were analysed 16 weeks after transplantation. **f** Flow cytometric analysis of the indicated cells from peripheral blood of donor mice. Ratios of CD45.1$^+$CD45.2$^+$ to CD45.2$^+$ T cells, NK cells and granulocytes in chimeras were analysed and shown as means ± S.D. **$P < 0.01$. ($n = 6$). Student's $t$-test was used as statistical analysis. Data represent at least three independent experiments

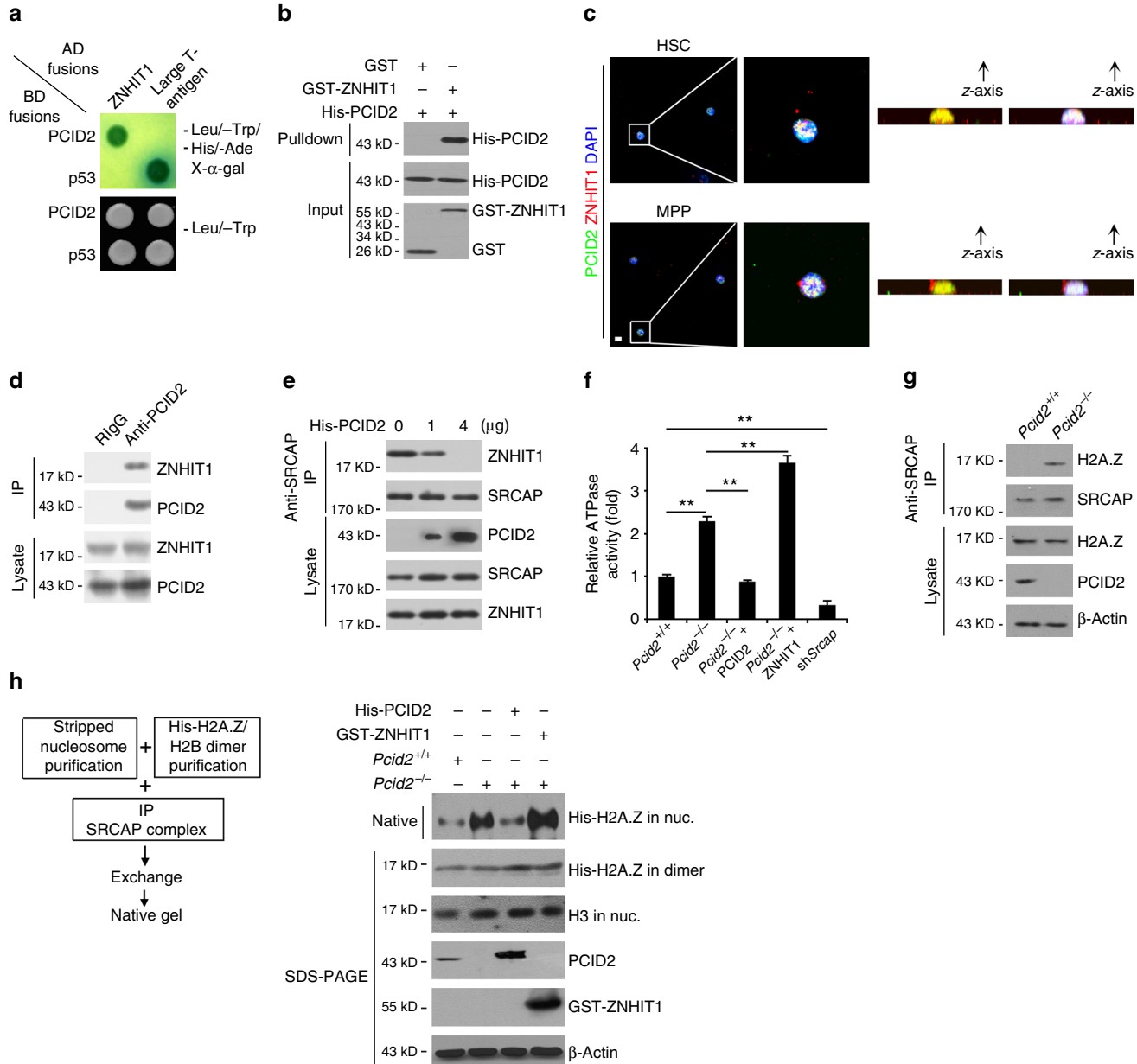

**Fig. 4** PCID2 interacts with ZNHIT1 to block SRCAP complex activity causing prevention of H2A/H2A.Z exchange. **a** PCID2 associates with ZNHIT1 by yeast two-hybrid screen. Yeast strain AH109 was co-transfected with Gal4 DNA-binding domain (BD) fused PCID2 and Gal4 activating domain (AD) fused ZNHIT1. Interaction of BD-p53 and AD-large T antigen was used as a positive control. **b** GST-ZNHIT1 was incubated with His-PCID2 protein followed by pulldown assays. **c** Mouse BM HSCs and MPPs were sorted, fixed and co-stained with the indicated antibodies. PCID2, green; ZNHIT1, red; nucleus, blue. Scale bar, 10 μm. Z-stack projection was performed to show nuclear co-localization. More than 50 cells of each group were analysed in three individual experiments. **d** LSK lysates were incubated with anti-PCID2 antibody. PCID2-associated ZNHIT1 was detected by immunoblotting. IP immunoprecipitation, RIgG rabbit IgG. **e** Recombinant PCID2 protein inhibited the binding of ZNHIT1 with SRCAP. MPP lysates were incubated with anti-SRCAP antibody and protein A/G beads in 4 °C overnight. Recombinant PCID2 protein was added with different concentrations, followed by anti-SRCAP antibody precipitation. **f** Srcap silenced MPPs (shSrcap) were sorted from recipient mice transplanted with shRNA-retrovirus infected HSCs. MPP lysates were immunoprecipitated by anti-SRCAP antibody, followed by detection of ATPase activities. Mouse IgG IP was used as a background control. Purified His-tagged PCID2 or GST-tagged ZNHIT1 protein (4 μg for each protein) was added into the ATPase reaction buffer. Relative O.D. values were normalized with background controls. Fold changes were shown as means ± S.D. **$P < 0.01$. ($n = 6$). Student's t-test was used as statistical analysis. **g** MPP lysates were immunoprecipitated with anti-SRCAP antibody followed by immunoblotting with the indicated antibodies. **h** Assessment of histone exchange activity in vitro. Data represent at least three independent experiments

CD45.1/CD45.2-recipient mice (Fig. 3e and Supplementary Fig. 4D). Sixteen weeks after transplantation, CD45.2$^+$ T cells and NK cells derived from Pcid2$^{-/-}$ LT-HSCs were predominant in peripheral blood but decreased the numbers of granulocytes (Fig. 3f). Collectively, Pcid2 functions as a cell intrinsic factor that regulates lineage commitment of MPPs.

**PCID2 interacts with ZNHIT1 to block SRCAP complex activity.** To further explore the molecular mechanism that Pcid2 regulated lineage specification, we screened a mouse BM cDNA library using Pcid2 as bait by a yeast two-hybrid approach. Of the 57 positive clones we screened out, 13 clones were identified to be ZNHIT1. ZNHIT1 was further validated to be a interacting

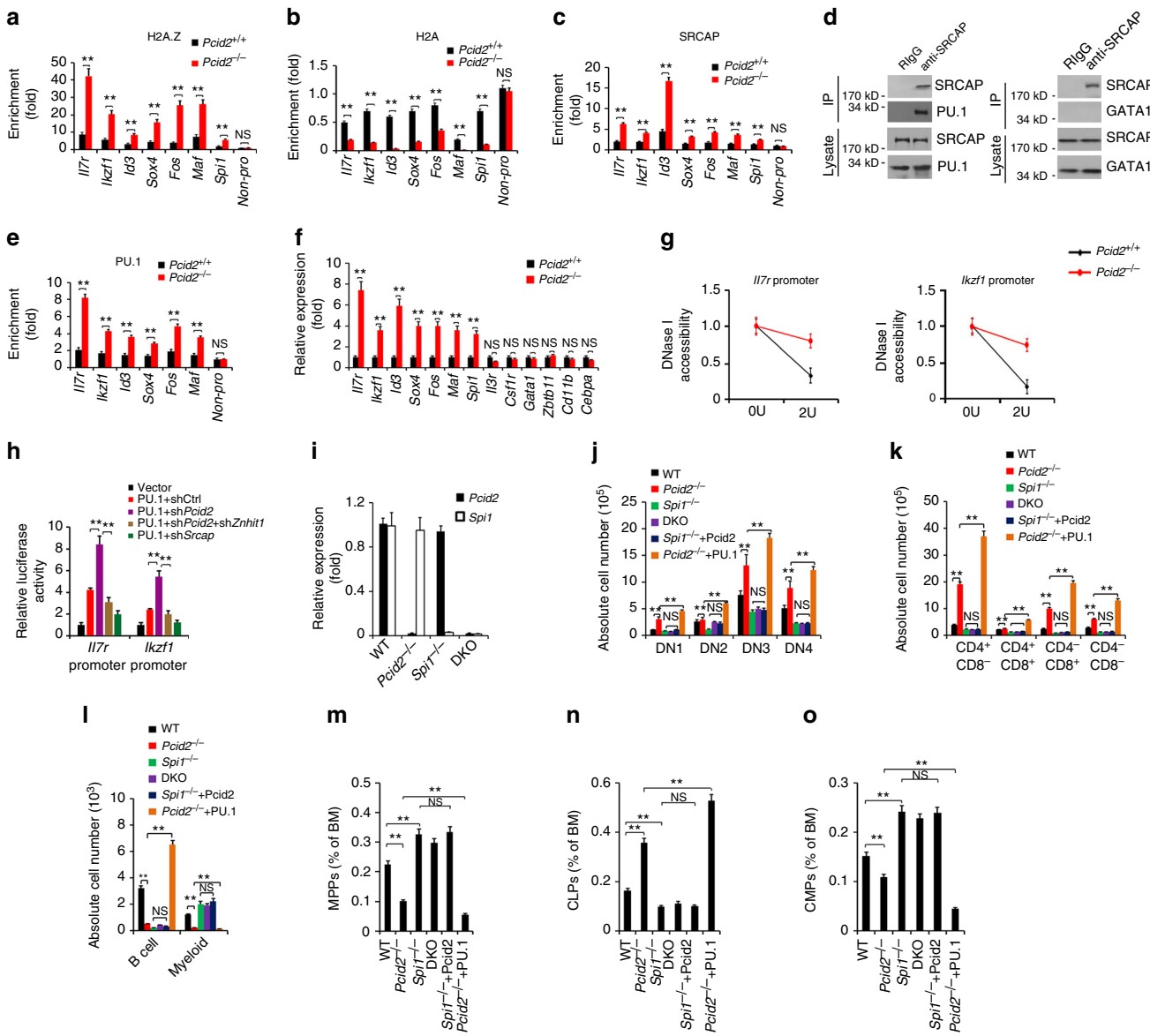

**Fig. 5** *Pcid2* deficiency causes H2A.Z deposition to lymphoid fate regulator genes in MPPs. **a** Sorted MPPs were lysed for ChIP assays. Indicated promoters were examined by qPCR. Signals were normalized to input DNA. Fold enrichment was calculated comparing with negative control (Non-pro locus). Results are shown as means ± S.D. **b, c** Indicated MPPs were lysed for ChIP assays with anti-H2A **b** or anti-SRCAP **c** antibody as in **a**. **d** MPP lysates were incubated with anti-SRCAP antibody. SRCAP-associated PU.1 and GATA1 were detected by immunoblotting. IP immunoprecipitation, RIgG rabbit IgG. **e** Indicated MPPs were lysed for ChIP assays with anti-PU.1 antibody as in **a**. **f** Expression levels of lymphoid fate regulator genes and myeloid fate regulator genes were assessed in sorted MPPs by quantitative RT-PCR. **g** Nuclei of MPPs were extracted for DNase I digestion assay. Chromatin accessibility were quantitated by qPCR. **h** Flag-PU.1, pTK and pGL3-*Il7r* promoter or pGL3-*Ikzf1* promoter, together with the indicated shRNAs, were transfected into 293T cells for luciferase assays. Results are shown as means ± S.D. **i** *Pcid2* and *Spi1* DKO were generated as described in Methods section. Bone marrow mRNA levels of *Pcid2* and *Spi1* were measured by qPCR. **j** In vitro lymphoid differentiation assays of the indicated LMPPs were conducted as in Fig. 2c. For Pcid2 or PU.1 restoration, $1 \times 10^3$ HSCs were isolated from the BM of *Spi1*$^{-/-}$ or *Pcid2*$^{-/-}$ mice and infected with pMY-GFP-Pcid2 or pMY-GFP-PU.1 containing retrovirus and transplanted into lethally irradiated recipient mice together with $5 \times 10^6$ helper cells. **k** Cells were sorted, infected and cultured as in **j** and used Flt3 ligand alone thereafter. **l** B-cell and myeloid differentiation assays of the indicated MPPs in vitro were conducted as in Fig. 2e. **m** Indicated HSCs as described in **j** were sorted, infected and transplanted for 16 weeks. **m** Percentages of MPPs, **n** CLPs and **o** CMPs from the BM of indicated mice were analysed by flow cytometry. Results are shown as means ± S.D. $n = 6$ for each group. **$P < 0.01$. Student's *t*-test was used as statistical analysis. QPCR primer pairs are shown in Supplementary Table 1. Data are representative of at least three independent experiments

protein of PCID2 (Fig. 4a). ZNHIT1, also called as p18^hamlet, is a main regulatory component of the SRCAP chromatin remodelling complex. Their direct interaction was further confirmed by a glutathione S-transferase (GST) pulldown assay (Fig. 4b). Moreover, ZNHIT1 was co-localized with PCID2 in the nucleus of MPPs (Fig. 4c). Anti-PCID2 antibody could precipitate ZNHIT1 from cell lysates of LSKs, suggesting that

PCID2 interacts with ZNHIT1 at endogenous expression levels (Fig. 4d) These data indicate that PCID2 associates with ZNHIT1 in MPPs.

Of note, increased doses of PCID2 could block the interaction of ZNHIT1 with SRCAP (Fig. 4e), suggesting that PCID2 protein inhibited the binding of ZNHIT1 with SRCAP. The SRCAP chromatin remodelling complex provides ATPase activity to replace

the H2A-H2B dimer of nucleosomes to the H2A.Z-H2B dimer for promotion of gene expression. We observed that *Pcid2* deficiency significantly ($P < 0.01$, *t*-test) augmented ATPase activity (Fig. 4f). Addition of PCID2 protein to *Pcid2*[−/−] MPP lysates could suppress ATPase activity to a comparable level of *Pcid2*[+/+] MPP lysates (Fig. 4f). Importantly, addition of ZNHIT1 protein to *Pcid2*[−/−] MPP lysates substantially enhanced ATPase activity (Fig. 4f). Expectedly, SRCAP depletion impaired ATPase activity. More importantly, *Pcid2* deficiency could enrich H2A.Z to the SRCAP complex in MPPs, whereas the SRCAP complex in *Pcid2*[+/+] MPPs did not associate with H2A.Z (Fig. 4g). Finally, we incubated purified mononucleosomes and recombinant His-H2A.Z-H2B dimers with purified SRCAP complexes from treated cells for an in vitro H2A.Z exchange assay (Fig. 4h). We observed that *Pcid2*-deficient MPPs caused substantial amounts of H2A.Z exchange, whereas addition of PCID2 protein in *Pcid2*[−/−] MPP lysates really blocked the H2A.Z exchange (Fig. 4h). Taken together, PCID2 binds ZNHIT1 to block the SRCAP remodelling activity, causing impairment of the H2A/H2A.Z exchange in MPPs.

**Pcid2 KO causes H2A.Z deposition to lymphoid fate genes.** We next sought to explore how *Pcid2* deficiency-mediated H2A.Z/H2A exchange drove MPPs toward lymphoid lineage commitment. We sorted MPPs from *Pcid2*[−/−] and *Pcid2*[+/+] mice and lysed them for chromatin immunoprecipitation (ChIP) assays with anti-H2A.Z and anti-H2A antibodies. We noticed that *Pcid2* deletion enriched more H2A.Z deposition to the promoters of lymphoid fate regulator genes (Fig. 5a). In contrast, *Pcid2* KO almost lost H2A deposition to these promoters of lymphoid fate regulator genes (Fig. 5b). Consistently, the promoters of lymphoid fate regulator genes in *Pcid2*[−/−] MPPs bound substantial amounts of SRCAP (Fig. 5c). Of these lymphoid fate regulator genes, the Ets transcription factor PU.1 (encoded by *Spi1* gene) is indispensable for the primary expression of lymphoid fate regulator transcripts for lymphocyte fate determination[27]. By contrast, GATA1 is a critical transcription factor for myelopoiesis[28]. Of note, PU.1 could associate with SRCAP by co-IP assay, while GATA1 did not interact with SRCAP in MPPs (Fig. 5d). Consistently, in *Pcid2*[−/−] MPPs, PU.1 was enriched at the promoters of lymphoid fate regulator genes (Fig. 5e). By contrast, *Pcid2* deletion did not cause H2A.Z, SRCAP and PU.1 deposition onto the promoters of myeloid lineage effector genes (Supplementary Fig. 5A). These data suggest that PU.1 behaves differently in the loci of lymphoid and myeloid lineage effector genes. Consequently, lymphoid fate regulator genes were highly expressed in *Pcid2*[−/−] MPPs, whereas myeloid fate regulator genes were not significantly ($P > 0.05$, *t*-test) changed in *Pcid2*[−/−] MPPs compared to *Pcid2*[+/+] MPPs (Fig. 5f).

In addition, *Pcid2* deficiency augmented chromatin accessibility to DNase I digestion at the promoters of lymphoid fate regulator genes, such as *Il7r* and *Ikzf1* (encoding Ikaros) (Fig. 5g). We next performed luciferase assays to confirm our observations. We found that *Pcid2* depletion remarkably promoted *Il7r* transcription (Fig. 5h). However, depletion of *Znhit1* or *Srcap* could dramatically suppress *Il7r* expression (Fig. 5h). Parallelly, an *Ikzf1* promoter reporter luciferase assay obtained similar results (Fig. 5h). We next performed fragment mapping on the *Il7r* promoter-enhancer loci. We observed that PU.1, SRCAP and ZNHIT1 were co-occupied at the same site of *Il7r* promoter-enhancer locus (−1400 to −1200 nt), whereas PCID2 did not bind to the same locus (Supplementary Fig. 5B). The binding of ZNHIT1 to SRCAP prevented ZNHIT1 from binding to PCID2. Meanwhile, we used the myeloid differentiation gene *Csf1r* as an assay control. *Csf1r* harbours a PU.1-binding site on the promoter region[29]. We noticed that PU.1 was accumulated at the position

of −200 to 0 nt of the *Csf1r* promoter (Supplementary Fig. 5B). However, SRCAP and ZNHIT1 were not co-occupied at the same locus on *Csf1r* promoter-enhancer region (Supplementary Fig. 5B). Furthermore, depletion of *Pcid2*, *Znhit1* or *Srcap* could not suppress *Csf1r* transcription (Supplementary Fig. 5C). These data suggest that *Pcid2* deficiency causes H2A.Z and SRCAP deposition to lymphoid fate regulator genes in MPPs to drive lymphoid lineage specification.

PU.1 deletion blocks lymphoid lineage commitment and promotes granulopoiesis[30]. We then established *Pcid2*[−/−]*Spi1*[−/−] double KO (DKO) mice. *Pcid2*[flox/+];*Spi1*[flox/+] mice were crossed with MxCre[+] mice to generate *Pcid2*[flox/flox];*Spi1*[flox/flox];MxCre[+] mice, followed by intraperitoneal injection with 200 μg poly(I:C) for conditional deletion of *Pcid2* and *Spi1* in haematopoietic system (Fig. 5i). We isolated LMPPs from the BM of *Pcid2*[−/−] mice, *Spi1*[−/−] mice or *Pcid2*[−/−]*Spi1*[−/−] DKO mice and cultured them with stromal OP9-DL1 cells for in vitro lymphoid lineage differentiation assays. We noticed that DKO LMPPs exhibited similar lymphoid lineage commitment tendency compared to *Spi1*[−/−] LMPPs (Fig. 5j, k). By contrast, *Spi1*[−/−] LMPPs failed to differentiate into CD44[−]CD25[−] cells, whereas *Pcid2*[−/−] LMPPs differentiated into more CD44[−CD25−] cells as previously demonstrated (Fig. 5j, k). Importantly, overexpression of Pcid2 in *Spi1*[−/−] LMPPs displayed a comparable ratio of lymphoid lineage commitment compared to *Spi1*[−/−] LMPPs alone (Fig. 5j, k and Supplementary Fig. 5D, E). However, overexpression of PU.1 in *Pcid2*[−/−] [L]MPPs significantly ($P < 0.01$, *t*-test) promoted lymphoid lineage commitment compared with *Pcid2*[−/−]LMPPs (Fig. 5j, k and Supplementary Fig. 5E). In addition, sorted MPPs were co-cultured with OP9 stromal cells to induce B-cell and myeloid cell differentiation. As expected, *Pcid2*[−/−] MPPs produced fewer B cells and myeloid cells compared with *Pcid2*[+/+] MPPs, whereas *Spi1*[−/−] MPPs tended to differentiate into myeloid lineage (Fig. 5l and Supplementary Fig. 5E). In contrast, DKO MPPs exhibited similar B-cell and myeloid lineage commitment tendency compared to *Spi1*[−/−] MPPs (Fig. 5l). However, overexpression of PU.1 in *Pcid2*[−/−] MPPs significantly ($P < 0.01$, *t*-test) promoted B cells but inhibited myeloid lineage commitment compared with *Pcid2*[−/−] MPPs (Fig. 5l, and Supplementary Fig. 5D). Similar observations were validated by in vivo BM transplantation assays (Fig. 5m–o and Supplementary Fig. 5F). These results indicate that Pcid2 acts as an upstream factor of PU.1 to regulate lymphoid lineage commitment.

**Znhit1 deletion abrogates H2A/H2A.Z exchange.** To explore the role of ZNHIT1 in the regulation of lymphoid lineage commitment, we deleted *Znhit1* gene in HSCs. We generated *Rosa26-LSL-Cas9*[+];*Vav-Cre*[+] mice by crossing B6;129-Gt(ROSA)26Sor[tm1(CAG-cas9*,−EGFP)Fezh]/J knockin mice with Vav-Cre transgenic mice and sorted their HSCs for sgRNA against *Znhit1* lentivirus-mediated genome editing[31], followed by in vivo transplantation. After successfully reconstituting the recipient BM, we sorted MPPs from these mice for further studies. As expected, *Znhit1* was completely deleted in MPPs (Fig. 6a). Of note, *Znhit1*[−/−] MPPs abrogated H2A.Z deposition to lymphoid fate regulator genes (Fig. 6b). Consequently, *Znhit1*[−/−] MPPs inhibited the lymphoid lineage differentiation but induced myeloid lineage differentiation via in vitro differentiation assays (Fig. 6c–e and Supplementary Fig. 6A–C). Finally, we next analysed donor-derived BM haematopoietic progenitor cells and peripheral blood cells in *Znhit1*[−/−] HSCs reconstituted recipient mice. We observed that *Znhit1* deficiency led to reduced BM cellularity and a decreased number of CLPs but an increased number of CMPs compared with WT mice (Fig. 6f, g). In parallel, *Znhit1* deficiency caused a shifted myeloid lineage specification

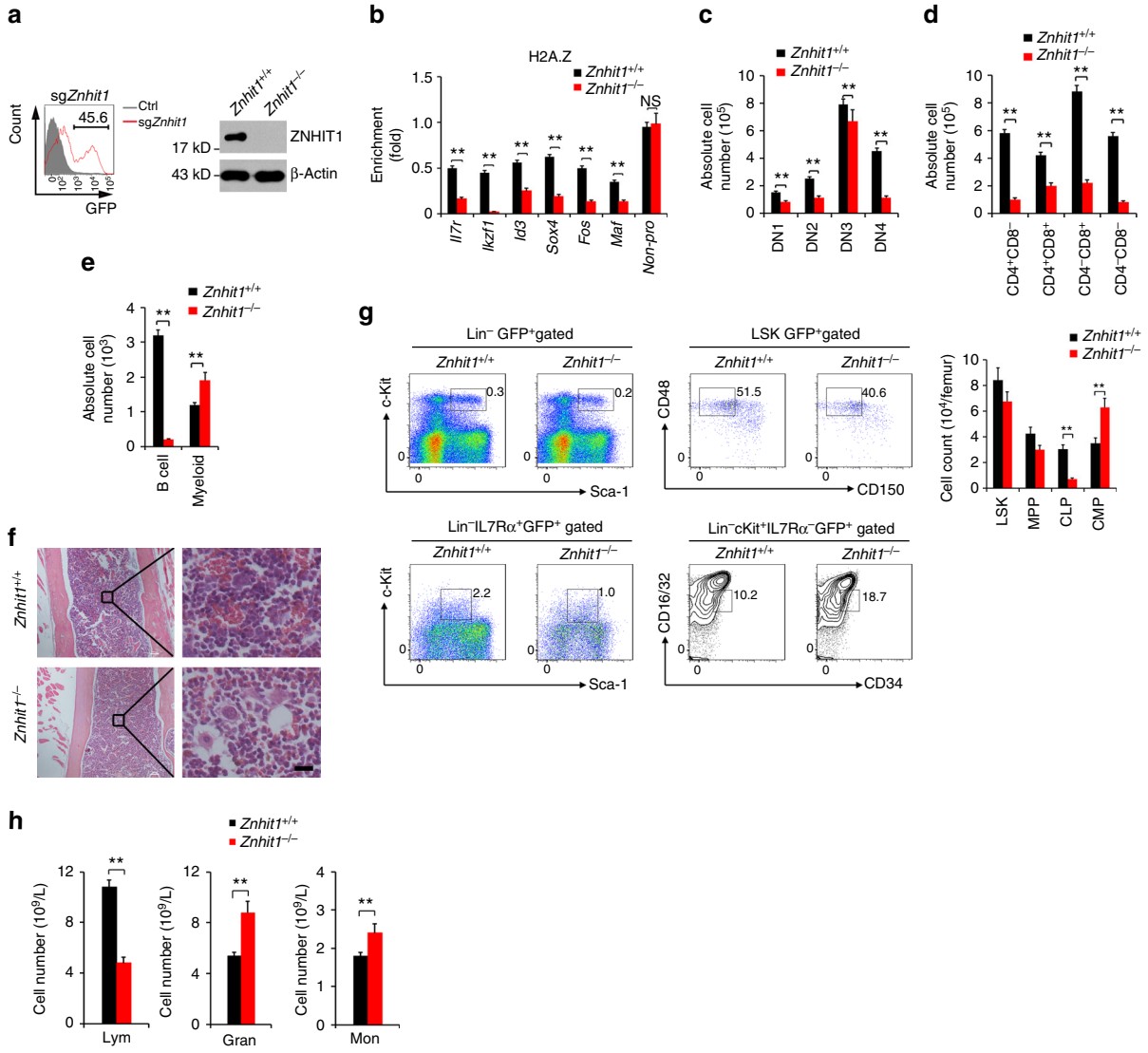

**Fig. 6** *Znhit1* deletion abrogates H2A/H2A.Z exchange to disrupt skewed lymphoid lineage commitment. **a** B6;129-*Gt(ROSA)26Sor*[tm1(CAG-cas9*,−EGFP)Fezh]/J knockin mice were crossed with Vav-Cre mice to obtain tissue-specific Cas9 expression. HSCs were sorted and infected with sg*Znhit1*-containing lentivirus and then transplanted into lethally irradiated recipient mice (CD45.1) for 8 weeks. ZNHIT1 was detected by immunoblotting. β-Actin were used as a loading control. **b** Indicated MPPs were sorted and lysed for ChIP assays with anti-H2A.Z antibody. Indicated promoters were examined by real-time qPCR. Signals were normalized to input DNA. **c** Indicated LMPPs were sorted and co-cultured with OP9-DL1 stromal cells as in Fig. 2c. Data are representative of six independent transfection and transplantation mice for each group. **P < 0.01. **d** Cells were sorted, infected and cultured as in **c** and used Flt3 ligand alone thereafter. After 4 days, cells were harvested and analysed by flow cytometry. n = 6 for each group. **P < 0.01. **e** Indicated MPPs were sorted and co-cultured with OP9 stromal cells as in Fig. 2c. Data are representative of six independent transfection and transplantation mice for each group. **P < 0.01. **f** Paraffin sections from the femurs of *Znhit1*-deficient mice were stained with H&E. Scale bars, 20 μm. **g** Flow cytometric analysis of LSKs, MPPs, CLPs and CMPs from the BM of the indicated mice. (n = 5); **P < 0.01. **h** Peripheral blood cell numbers were analysed from the indicated transplanted recipient mice. Lymphocytes (Lym), monocytes (Mon), and granulocytes (Gran) were analysed. Results are shown as means ± S.D. Data are representative of six independent transfection and transplantation mice for each group. **P < 0.01. Student's *t*-test was used as statistical analysis

(Fig. 6h). Therefore, *Znhit1* deficiency results in an opposite phenotype vs. *Pcid2* KO mice. Collectively, *Znhit1* deletion abrogates H2A/H2A.Z exchange in MPPs to disrupt lymphoid lineage commitment capacity.

## Discussion

Lymphoid lineage commitment is an important process in haematopoiesis, which forms an immune system to protect a host from pathogen invasion. Within the BM, haematopoietic progenitors undergo lineage commitment towards a given lineage via the loss of ability to generate other lineages[32]. With lineage commitment signals, MPPs, differentiating from short-term

HSCs (ST-HSC), give rise to two different progenitors CLPs and CMPs[1,33], causing generation of mature lymphoid and myeloid cells in periphery, respectively. However, how MPPs switch into CLPs or CMPs has not been defined yet. In this study, we show that Pcid2 is highly expressed in the BM and MPPs. *Pcid2* deletion in the haematopoietic system causes skewed lymphoid lineage specification. *Pcid2*-deficient HSCs prefer to differentiate into lymphoid cells but not into myeloid cells. In MPPs, PCID2 interacts with ZNHIT1 to block the SRCAP complex activity and prevents the H2A/H2A.Z exchange at nucleosomes of key lymphoid fate regulator genes. Pcid2 acts as one of the upstream inhibitors of PU.1 to impede the expression of key lymphoid fate regulator genes. Finally, *Znhit1* deletion abrogates

H2A/H2A.Z exchange in MPPs to disrupt lymphoid lineage commitment capacity (Supplementary Fig. 6D).

Upon sensing differentiation signals, CLPs generate all lymphoid lineages but not any myeloid cells[7,26], whereas its counterpart CMPs produce all myeloid cell types[33,34], supporting that the lymphoid and myeloid developmental programmes independently operate downstream of HSCs. Many reports showed that extrinsic cytokine signals can direct lineage-restricted capacities on HSCs. Erythropoietin (Epo) initiates erythroid lineage bias at all lineage bifurcations between HSCs and erythroid progenitors. We demonstrated that extrinsic insulin signalling drives lymphoid lineage commitment in early lymphopoiesis[35]. Here we show that HSCs from Pcid2-deficient mice cause decreased MPPs and CMPs but increased CLPs in BM, causing skewed lymphoid lineage specification. The decreased numbers of MPPs and CMPs might be caused by dysregulated proliferation or cell death. Actually, we noticed that Pcid2-deficient MPPs and CMPs exhibit a similar cycling ratio to WT mice and have no apparent apoptosis. Through in vivo engraftment, we validate that Pcid2 acts as a cell intrinsic factor to restrict lymphoid lineage specification.

Pcid2 was originally identified as a partner to export mRNAs from the nucleus to cytoplasm[11]. PCID2 can interact with the repair factor BRCA2 of the human TREX-2 complex to prevent RNA-mediated genome instability[12]. Pcid2 also participates in the regulation of self-renewal of mouse ECSs[13]. We showed that PCID2 is in the CBP/p300-EID1 complex to sustain the pluripotency of human and mouse ESCs via regulation of EID1 stability[14]. In this study, we show that Pcid2 deficiency fails to impact the number of HSCs (Fig. 1g and Supplementary Fig. 2B). These data suggest Pcid2 is not implicated in the regulation of self-renewal of HSCs. We demonstrate that PCID2 interacts with ZNHIT1, a master regulatory subunit of SRCAP complex, to replace ZNHIT1 out from the SRCAP complex of the nucleosomes of lymphoid fate regulator genes in MPPs. Replacement of ZNHIT1 by PCID2 causes inactivation of ATPase activity of the SRCAP complex to block H2A/H2A.Z exchange, causing suppression of lymphoid fate regulator gene expression.

Epigenetic modulations are involved in a variety of biological processes, including gene transcription, DNA replication, DNA repair and DNA recombination[36]. These processes include post-translational modulations of histones, DNA methylation, incorporation of histone variants and nucleosome remodelling activity. Of note, the nucleosome remodelling and incorporation of histone variants largely rely on assistance of ATP-dependent chromatin remodelling complexes[37]. The chromatin remodelling complexes have been reported to participate in the maintenance of HSC pluripotency[38]. BAF53a, a subunit of the SWI/SNF-like BAF complex, is implicated in the survival and maintenance of HSCs[18]. SNF2-like ATPase mi-2β subunit of the nucleosome remodelling deactylase (NuRD) complex is required for the maintenance of HSCs[39]. We previously showed that WASH protein associates with the nucleosome remodelling factor (NURF) complex to trigger c-Myc expression, which directs the differentiation commitment of HSCs[40]. However, how the chromatin remodelling complexes regulate lymphoid and myeloid branchpoint lineage specification still remains elusive. Here we show that Pcid2 deficiency causes the SRCAP complex enrichment at the lymphoid fate regulator genes to drive H2A.Z exchange for these gene expression, resulting in shifted lymphoid lineage differentiation. ZNHIT1, also called as p18[Hamlet], is a master regulatory subunit of the SRCAP complex[41]. The SRCAP complex harbors ATPase activity to supply ATP for exchange of H2A to the variant H2A.Z at nucleosomes[42–44], ensuring the loose chromatin structure necessary for transcriptional activation. Mutations of SRCAP are related to the pathogenesis of Floating-

Harbor syndrome[45], suggesting the SRCAP complex has a critical function in physiological and pathological processes. The SRCAP complex causes H2A.Z exchange at the muscle-specific gene promoters to direct muscle differentiation[21]. Here we show that Znhit1 deletion in the haematopoietic system abrogates H2A.Z exchange at lymphoid fate regulator gene promoters to impede the shifted lymphoid lineage commitment.

Accumulating evidence supports that combinations of transcription factors coordinately and sequentially modulate lymphopoiesis. For example, PU.1 (encoded by Spi1 gene) and Ikaros (encoded by Ikzf1 gene) are expressed in earlier haematopoietic progenitors that take part in the regulation of lymphoid lineage commitment[46,47]. Neonatal Ikzf1-null mice display complete defect in fetal thymocyte development[33]. Adult Ikzf1-deficient mice impair lymphoid differentiation and exhibit thymocyte development shifted to CD4 T cells[48]. PU.1 is one of the most important regulators of lymphoid and granulocyte/monocyte (GM) lineage differentiation[49]. PU.1 is also involved in the generation of early GM and lymphoid progenitors such as CMPs, GMPs and CLPs[27]. Interestingly, PU.1+ MPPs exhibit granulocyte/monocyte/lymphoid-restricted progenitor activity without megakaryocyte/erythroid differentiation capacity[28], while GATA-1+ MPPs display potent myeloerythroid potential without producing lymphocytes. Furthermore, PU.1 and GATA-1 mutually suppress each other's expression and transactivation functions[50,51], suggesting that the reciprocal activation of PU.1 and GATA-1 primarily organizes the haematopoietic lineage fate decision to generate the earliest lymphoid or myeloid haematopoietic progenitors.

Besides the antagonistic relationship with GATA-1, PU.1 acts also by its expression levels. High levels of PU.1 promote myeloid development, while low levels drive B-cell development[52]. In this study, we show that Pcid2 acts as an upstream regulator of PU.1 to suppress its activity in MPPs, resulting in restrict lymphoid lineage specification. In the presence of PCID2, it binds to ZNHIT1 to block the SRCAP complex assembly causing prevention of H2A/H2A.Z exchange on Spi1 promoter, which deposits H2A to suppress its expression. In the absence of PCID2, freed ZNHIT1 assembles the SRCAP complex to enrich H2A.Z onto the Spi1 promoter, which initiates its expression to drive lymphoid lineage commitment. Our results demonstrate that PU.1 expression in MPPs promote lymphoid lineage specification via Pcid2-mediated H2A/H2A.Z exchange on Spi1 promoter. Whether differing concentrations of PU.1 affects MPPs to lymphoid lineage specification remains to be further studied.

It has been reported that other factors such as Gfi-1 and Egr proteins are also involved in the modulation of myeloid and lymphoid lineage decisions. Gfi-1 is implicated in the regulation of lymphocyte development and activation[53]. Gfi-1 deficiency causes defects in B- and T-cell differentiation[53]. Moreover, Gfi-1 also takes part in regulating homeostasis of HSCs by coordinating proliferation and migration[54]. Gfi-1 also directly associates with and suppresses transactivation by PU.1 in favouring granulopoiesis[55]. In addition, PU.1 also induces the expression of Egr proteins. Egr-1 and Egr-2 are zinc-finger transcription factors that are able to activate or repress transcription. Gene activations induced by Egr-1 or Egr-2 apparently promotes monocytic maturation[55]. However, how these factors tightly coordinate in the regulation of myeloid and lymphoid lineage decisions still need to be further investigated.

Here we show that DKO Pcid2 and Spi1 genes display a similar ratio of lymphoid vs. myeloid progenitors and mature cells in comparison with Spi1−/− mice. Pcid2 acts as an upstream regulator of PU.1 to suppress its activity in MPPs, causing restrict lymphoid lineage specification. In addition, PCID2 binds to ZNHIT1 to impair the SRCAP complex activity that suppresses the expression of lymphoid fate regulator genes. Therefore, we

conclude that Pcid2 has a critical function in the earliest hae-matopoietic branchpoint specification that directs the myeloid and lymphoid progenitor populations.

## Methods

**Cell culture.** Human 293T cells (ATCC, CRL-3216) were cultured with Dulbecco's modified Eagle's medium supplemented with 10% fetal bovine serum (FBS) and 100 U/ml penicillin and 100 mg/ml streptomycin. Retrovirus and lentivirus infecting primary BM cells were produced in 293T cells by using the standard protocols. Transfection was performed using lipofectin reagent (Invitrogen). For in vitro differentiation, MPPs were plated on monolayer of OP9-DL1 in α-Minimum Essential Medium containing 20% FBS, 100 μg/ml streptomycin and 100 U/ml penicillin and supplemented with 1 ng/ml rmIL-7 and 5 ng/ml rmFlt3L. MPPs were also sorted and cultured on OP9 feeder cells (ATCC, CRL-2749) supplemented with recombinant mouse SCF (10 ng/ml), recombinant mouse Flt3-ligand (20 ng/ml) and recombinant mouse IL-7 (1 ng/ml). Cell lines were obtained from ATCC and authenticated by PCR. Mycoplasma contamination had been tested by PCR and excluded.

**Antibodies and reagents.** A rabbit polyclonal antibody against human PCID2 specifically recognized mouse and human PCID2 was raised using purified His-PCID2 protein. PerCP-CY5.5-anti-CD127(A7R34), Alexa Fluor 700-anti-CD34 (RAM34), APC-anti-c-Kit (2B8), fluorescein isothiocyanate (FITC)-anti-Sca-1 (D7), PE-CY7-anti-CD16/32 (93), eFluor450 lineage detection cocktail (Cat# 88-7772-72), PE-Cy7-anti-CD150 (mShad150), APC-Cy7-anti-CD48 (HM48-1), PerCP-Cy5.5-anti-Thy-1 (HIS51), APC-anti-CD44 (IM7), PE-CD122 (5H4), FITC-Anti--NK1.1 (PK136), APC-anti-CD49b (DX5), PE-anti mouse IgM (eB121-15F9), FITC-anti-CD43 (eBioR2/60), PE-anti-F4/80 (BM8), FITC-Anti-CD11b (M1/70), APC-anti-Gr1 (RB6-8C5), PE-CY5-anti-CD3 (SK7), PE-anti-CD4 (GK1.5), FITC-anti-CD8 (SK1), FITC-anti-BP-1 (6C3), FITC-anti-CD244 (2B4) and PE-anti-B220 (2D1) antibodies were purchased from eBiosciences (San Diego, USA). The antibody against mouse CD34 (581/CD34) was from BD Biosciences (Franklin Lakes, USA). The antibody against mouse CD229 (Ly9ab3) was from Biolegend (San Diego, USA). These antibodies were used in a 1:100 dilution for flow cytometric staining. Antibodies against ZNHIT1 (Cat# sc-427622) and GST-tag (Cat# sc-33613) were purchased from Santa Cruz Biotechnology (Santa Cruz, USA). Antibodies against β-actin (AC-74) and His-tag (HIS-1) were from Sigma-Aldrich (St. Louis, USA). These antibodies were used in a 1:5000 dilution for western blotting. The antibody against SRCAP was from Kerafast (Cat# ESL104; Boston, USA). Antibodies against H2A.Z(ab4626) and Ki67 (ab15580) were purchased from Abcam (Cambridge, USA). Anti-PU.1 antibody was purchased from Thermo fisher scientific (Cat# PA5-17505; Rockford, USA). Secondary antibodies conjugated with Alexa-594, Alexa-488, Alexa-405 or Alexa-649 were purchased from Molecular probes Inc (Eugene, USA). Hoechst 33342 were purchased from Sigma-Aldrich. His-bound resin was from Millipore. GST-sepharose was from GE Healthcare (Buckinghamshire, UK). SuperReal premix plus qPCR buffer was from TIANGEN Biotech (FP205, Beijing, China). ATPase/GTPase activity assay kit was purchased from Sigma-Aldrich (MAK-113). Annexin V-FITC/propidium iodide (PI) apoptosis detection kit was purchased from BD Biosciences (556547, Franklin Lakes, USA). Poly (I:C) was purchased from Sigma-Aldrich.

**Mouse strains.** Mouse experiments were approved by the Institutional Animal Care and Use Committees at the Institute of Biophysics, Chinese Academy of Sciences. Pcid2$^{flox/flox}$ mice were generated by homologous recombination as described[15]. Spi$^{flox/flox}$ mice were generated by Dr. Daniel G. Tenen from Harvard University[27]. B6.Cg-Tg(Mx1-cre)1Cgn/J (Mx-Cre, JAX-003556), B6.Cg-Tg(Vav1-iCre)A2Kio/J (Vav- Cre, JAX-008610) transgenic mice and B6;129-Gt(ROSA)26Sor$^{tm1(CAG-cas9*,−EGFP)Fezh}$/J (JAX-024857) knockin mice were from Jackson Lab. For generation of Pcid2 deficient mice, Pcid2$^{flox/flox}$ mice were crossed with MxCre$^+$ mice to generate Pcid2$^{flox/flox}$;MxCre$^+$ mice. Pcid2$^{+/+}$;MxCre$^+$ mice were used as littermate control. Mice were then intraperitoneally injected with 200 μg polyinosine-polycytidylic acid (poly(I:C)) every other day for three times to induce Pcid2 deletion. Pcid2$^{flox/flox}$ mice were also crossed with Vav-Cre$^+$ mice to generate Pcid2$^{flox/flox}$;Vav-Cre$^+$ mice as Pcid2$^{−/−}$, whereas Pcid2$^{flox/flox}$;Vav-Cre$^-$ mice are referred to as Pcid2$^{+/+}$. For generation of Pcid2 and Spi1 DKO, Pcid2$^{flox/flox}$ mice were crossed with Spi1$^{flox/flox}$ mice firstly to obtain Pcid2$^{flox+}$;Spi1$^{flox/+}$mice. Then mice were crossed with MxCre$^+$ mice to generate Pcid2$^{flox/flox}$;Spi1$^{flox/flox}$;MxCre$^+$ mice. Then mice were intraperitoneally injected with 200 μg poly(I:C) every other day for three times to induce double deletion of Pcid2 and Spi1. Pcid2 and Spi1 deletion were confirmed by reverse transcriptase-PCR (RT-PCR) and immuno-blotting. For generation of Znhit1 deficiency in BM, B6;129-Gt(ROSA)26Sor$^{tm1(CAG-cas9*,−EGFP)Fezh}$/J knockin mice were crossed with Vav-Cre transgenic mice to generate Rosa26-LSL-Cas9$^+$;Vav-Cre$^+$ mice. HSCs were sorted and infected with lentiCRISPRv2 containing sgZnhit1 lentivirus. $1 \times 10^3$ HSCs together with $5 \times 10^6$ helper cells were then transplanted into lethally irradiated (10 Gy, $^{60}$Co irradiation) recipient mice (CD45.1). Donor-derived BM haematopoietic progenitor cells and peripheral blood cells in recipient mice were analysed 8 weeks after transplantation. Znhit1 deletion was confirmed by RT-PCR and immunoblotting. All the mice we used were of C57BL/6 background, female and 8–12-week old. We were not blinded to the group and did not use randomization in our animal studies.

Littermates with the same age and gender for each group were used. We excluded the mice 5 g thinner than other littermates before any treatment or analysis. Mice were bred in specific pathogen-free animal facility. The euthanasia method was cervical dislocation when needed.

**Plasmids.** RNAi sequences were designed according to BLOCK-IT RNAi Designer system instructions (Invitrogen). shRNA oligos encoding target sequences against Pcid2 (5′-GATCATCACCTACAGGAAC-3′), Znhit1 (5′-CCTGGAGAATGA-CAACTTC-3′), Srcap (5′-GAAAGAAATTACTGACATT-3′) or scramble sequence (5′-AATTCTCCGAACGTGTCACGT-3′) were cloned into MSCV-LTRmiR30-PIG vector (LMP, Openbiosystems). Mouse Pcid2 was cloned into p3×flag-CMV-9 expression vector. Mouse Znhit1 was cloned into pCDNA4-Myc-His expression vector. For recombinant protein purification, mouse Znhit1 was cloned into pGEX-6P-1 vector (GE Healthcare), expressed in Escherichia coli and purified using Glutathione Sepharose 4B beads according to the manufacturer's instruction. Mouse full-length Pcid2 was subcloned into pETDuet vector for E. coli recombinant expression and purification using Ni-NTA His-resins from Novagen according to the manufacturer's instruction. Mouse Spi1 and Pcid2 were also cloned into pMY-IRES-GFP vector for packaging retrovirus according to the manufacturer's instruction. sgZnhit1 (GCGGCGGATCAACCGACAGC) was cloned into lentiCRISPRv2 vector. Mouse Il7r promoter region and Ikzf1 promoter region were cloned into pGL4 basic vector (Promega) for luciferase assays[56].

**Histology.** Mouse thymus and cervical lymph nodes were fixed in 4% paraf-ormaldehyde (PFA) for 12 h. Fixed tissues were washed for twice using 75% ethanol and embedded in paraffin, followed by sectioning and staining with hae-matoxylin and eosin according to standard laboratory procedures. For BM his-tology analysis, femurs were fixed in 4% formaldehyde followed by decalcifying in 10% EDTA-PBS (phosphate-buffered saline) buffer. Longitudinal paraffin sections were prepared for haematoxylin and eosin staining. Mouse femurs were fixed in PFA–lysine–periodate buffer for 12 h, then decalcified in decalcifying buffer (10% EDTA in PBS (w/v), pH 7.4) for 48 h and then rehydrated in 30% sucrose solution for 48 h before snap frozen in OCT (TissueTek). Cytosections were obtained according to standard procedures[57].

**Analysis of peripheral blood cells.** Mice were anesthetized and peripheral blood was collected from the inferior vena cava with EDTA as anticoagulant. Blood was analysed by XFA6030 automated hemacytometer (Slpoo). Cell numbers and percen-tages of each population were counted. For morphological analysis, peripheral blood smears were stained by Wright's. Images were taken in oil-immersion 100× lens.

**Flow cytometry.** Mice were euthanized and BM cells were flushed out from femurs in PBS buffer. Spleen cells and thymus cells were obtained by mashing tissues in PBS buffer. Cells were sifted through 70 μm cell strainers after removing red blood cells by suspending cells in ammonium-based red cell lysis buffer. Peripheral blood mono-nuclear cells (PBMCs) were obtained by suspending in ammonium-based red cell lysis buffer to remove red blood cells. For BM flow cytometric analyses, LSK (Lin$^-$Sca-1$^+$c-Kit$^+$), MPP (Lin$^-$Sca-1$^+$c-Kit$^+$CD48$^+$CD150$^-$), LT-HSC (Lin$^-$Sca-1$^+$c-Kit$^+$CD150$^+$CD48$^-$), LMPP (Lin$^-$Sca-1$^+$c-Kit$^+$CD150$^-$Flt3$^+$), CLP (Lin$^-$CD127$^+$Sca-1$^{low}$c-Kit$^{low}$), CMP (Lin$^-$c-Kit$^+$Sca-1$^-$CD34$^+$CD16/32$^-$), GMP (Lin$^-$c-Kit$^+$Sca-1$^-$CD34$^+$CD16/32$^+$), MEP (Lin$^-$c-Kit$^+$Sca-1$^-$CD34$^-$CD16/32$^-$), pro T subtypes of DN1 (CD44$^+$CD25$^-$), DN2 (CD44$^+$CD25$^+$), DN3 (CD44$^-$CD25$^+$), DN4 (CD44$^-$CD25$^-$), pre-pro B (B220$^+$CD43$^+$BP-1$^-$HSA$^-$), pro B ((B220$^+$CD43$^+$BP-1$^-$HAS$^+$), NKP (Lin$^-$CD122$^+$NK1.1$^-$DX5$^-$), MPP-1 (CD150$^-$CD48$^-$CD229$^-$CD244$^-$LSK), MPP-2 (CD150$^-$CD48$^-$CD229$^+$CD244$^-$ LSK), MPP-3 (CD150$^-$CD48$^-$CD229$^+$CD244$^+$LSK), granulocyte (Gr1$^+$CD11b$^+$) and monocyte (F4/80$^+$CD11b$^+$) populations were ana-lysed or sorted with a FACSAria II instrument (BD Biosciences). For peripheral cell flow cytometric analyses, CD3$^+$ T cells, NK1.1$^+$ NK cells, CD11b$^+$F4/80$^+$ monocytes and CD11b$^+$Gr1$^+$ granulocytes in PBMCs, CD4$^+$CD8$^-$ T cells and CD4$^-$CD8$^+$ T cells in thymus were analysed or sorted with a FACSAria II instrument (BD Biosciences). For cell cycle analysis, BM cells were stained with the indicated surface marker antibodies followed by staining of Ki67 and Hoechst 33342. MPPs were gated as Lin$^-$Sca-1$^+$c-Kit$^+$CD48$^+$CD150$^-$. Differential stages of cell cycle were analysed. For apoptosis analysis, BM cells were stained with the indicated surface marker antibodies, followed by staining with PI/Annexin V or active caspase 3 antibody. Data were analysed using the FlowJo 7.6.1 software[40].

**In vitro differentiation assay.** In vitro lymphoid differentiation assay was described previously[26]. LMPPs were sorted and cultured for 14 days on OP9-DL1 feeder cells supplemented with recombinant mouse Flt3-ligand (5 ng/ml), and recombinant mouse IL-7 (1 ng/ml). Then cells were collected for staining with anti-CD44 and anti-CD25 antibodies. For mature T-cell differentiation, LMPPs were sorted and cultured for 14 days on OP9-DL1 feeder cells supplemented with recombinant mouse IL-7 (1 ng/ml) and recombinant mouse Flt3-ligand (5 ng/ml), followed by co-culturing with OP9-DL1 using Flt3 ligand (5 ng/ml) alone for 4 days. Cells were then harvested and stained with anti-CD4 and anti-CD8 anti-bodies. MPPs were also sorted and cultured on OP9 feeder cells supplemented with recombinant mouse SCF (10 ng/ml), recombinant mouse Flt3-ligand (20 ng/ml) and recombinant mouse IL-7 (1 ng/ml) for 10 days. Then cells were collected for

staining with anti-CD19 and anti-Mac1 antibodies. For in vitro myeloid differentiation, MPPs were sorted from BM and cultured in methocult medium (Stemcells) for CFC formation assay. CFU-GEMM, CFU-M, CFU-G, CFU-GM and BFU-E colonies were counted, respectively, according to the manufacturer's instruction.

**Immunofluorescence assay**. Cells were placed on 0.01% poly-L-lysine-treated coverslips and fixed with 4% PFA for 20 min at room temperature, followed by 0.5% Triton X-100 permeabilization and 5% donkey serum blocking. Cells were then incubated with appropriate primary antibodies at 4 °C overnight followed by incubation with corresponding fluorescence-conjugated secondary antibodies. Nuclei were stained with 4,6-diamidino-2-phenylindole (DAPI). Images were obtained with Olympus FV1200 laser scanning confocal microscopy (Olympus, Japan). Confocal sections were taken every 1 μm through the whole cell for z-stack projection. The software ImageJ was used for co-localization quantitation. For each experiment, at least 50 typical cells were observed. For BM histology analysis, mouse femur cytosections were blocked with 10% donkey serum, followed by incubation with appropriate primary antibodies and corresponding fluorescence-conjugated secondary antibodies. Nuclei were stained with DAPI. Cytosections were washed and dehydrated in gradient EtOH followed by rinsing in xylene and coversliping. For thymus histology analysis, thymus was fixed in 4% formaldehyde followed by paraffin embedding, sectioning and antibody staining. Images were obtained with Zeiss Axio scan.Z1 scanning system (Zeiss, Germany).

**Immunoprecipitation assay**. 293T cells were transfected with the indicated plasmids and harvested. Freshly isolated MPP cells were from femurs. Cells were lysed with ice-cold RIPA buffer (50 mM Tris-HCl (pH 7.4), 150 mM NaCl, 0.5% sodium desoxycholate, 0.1% SDS, 5 mM EDTA, 2 mM PMSF, 20 mg/ml aprotinin, 20 mg/ml leupeptin, 10 mg/ml pepstatin A, 150 mM benzamidine and 1% Nonidet P-40) for 1 h. Lysates were incubated with the indicated antibodies followed by immunoprecipitation with protein A/G agarose beads and immunoblotting. See Supplementary Fig. 7 for uncropped blots.

**Real-time qPCR**. Total RNAs from different mouse tissues and BM cell populations were extracted with the RNA Miniprep Kit (DP-419, Tiangen, China) according to the manufacturer's protocol. For relative gene expression analysis, 2 μg total RNA per aliquot was used for synthesizing cDNA using M-MLV reverse transcriptase (M1701, Promega, USA). Quantitative PCR (qPCR) analysis and data collection were performed by ABI 7300 qPCR system using the primer pairs as listed in Supplementary Table 1. DNA from each ChIP sample was normalized to its corresponding input sample.

**ATPase activity assay**. Freshly sorted MPP cells were lysed with low salt buffer (20 mM Hepes pH7.9, 1.5 mM MgCl$_2$, 20 mM KCl, 0.2 mM EDTA) followed by extraction of nuclear components in high salt buffer (20 mM Hepes pH7.9, 1.5 mM MgCl$_2$, 1.2 M KCl, 0.2 mM EDTA). SRCAP was immunoprecipitated by the indicated antibodies and protein A/G beads from nuclear extracts. ATPase activities were measured using the ATPase/GTPase Activity Assay Kit (MAK-113, Sigma-Aldrich) according to the manufacturer's instruction[44].

**Chromatin immunoprecipitation assay**. ChIP was performed according to a standard protocol (Upstate, USA). Sheared chromatin (sonicated to 200–500 bp) from cells fixed in 1% formaldehyde was incubated with 4 μg antibody overnight at 4 °C, followed by immunoprecipitation with salmon sperm DNA/protein agarose beads. After washing, elution and cross-link reversal, DNA from each ChIP sample and its corresponding input sample were purified and analysed using qPCR[58].

**DNase I accessibility assay**. Nuclei were isolated from MPPs using the Nuclei Isolating Kit (NUC101, Sigma-Aldrich) according to the manufacturer's protocol. Nuclei were resuspended in 200 μl of DNase I digestion buffer (1 mM EDTA, 0.1 mM EGTA, 5% sucrose, 1mM MgCl$_2$, 0.5 mM CaCl$_2$). Two equal aliquots of 100 μl nuclei were treated with the indicated units of DNase I (Sigma, USA) and incubated at 37 °C for 5 min. Reactions were stopped by 2× DNase I stop buffer (20 mM Tris ph 8.0, 4 mM EDTA, 2 mM EGTA). DNA was extracted and analysed by qPCR.

**Yeast two hybrid screening**. Yeast two-hybrid screening was performed using Matchmaker Gold Yeast Tow-Hybrid system (Clontech laboratories. Mountain View, USA) following the manufacturer's instruction. Briefly, PCID2 was cloned into pGBKT7 plasmid as BD-Pcid2 bait. Yeast AH109 cells were transfected with BD-PCID2 and plasmids containing human spleen cDNA library (Clontech). Clones were further identified by DNA sequencing.

**BM transplantation**. For BM transplantation, donor BM cells described above were injected to lethally irradiated (10 Gy) recipient mice. Eight weeks after transplantation, percentages of donor-derived BM cell populations were analysed by flow cytometry. For competitive transplantation, donor BM HSC cells from

Pcid2$^{+/+}$ mice and Pcid2$^{-/-}$ mice were sorted and 1:1 mixed. HSCs together with $5 \times 10^6$ helper cells were injected into lethally irradiated recipient mice. Sixteen weeks after transplantation, percentages of donor-derived peripheral blood cell populations were analysed by flow cytometry. Blood was obtained from mice tail vein and stained as described above[40].

**Histone exchange assay**. Nuclei were isolated from mouse BM cells using the Nuclei isolating Kit (NUC101, Sigma-Aldrich) according to the manufacturer's protocol, followed by micrococool MNase digestion. Gel filtration chromatography was performed by Superdex 200 10/300 GL (GE Healthcare) according to the manufacturer's instruction to isolate stripped nucleosome fractions. SRCAP was immunoprecipitated by anti-SRCAP antibody and protein A/G beads from nuclear extracts of Pcid2$^{+/+}$ or Pcid2$^{-/-}$ MPPs. His-tagged H2A.Z and H2B protein were purified from E. coli and assembled to H2A.Z/H2B dimer in vitro. Purified His-tagged PCID2 or GST-tagged ZNHIT1 protein (4 μg for each protein) was directly added into the exchange reaction buffer (70 mM NaCl, 10 mM Tris-HCl (pH8.0), 5 mM MgCl$_2$, 0.1 mg/ml BSA, 2 mM ATP and 1 mM DTT). Isolated mononucleosomes, immunoprecipitated SRCAP complex, purified proteins of PCID2 or ZNHIT1 and recombinant His-H2A.Z-H2B dimers were incubated at 37 °C for 30 min and resolved by native gel. His-tagged H2A.Z exchanging into nucleosomes was probed by immunoblotting[59].

**Statistical analysis**. Student's t-test was used as statistical analysis by using Microsoft Excel or SPSS 13.

**Data availability**. All data generated or analysed during this study are included in this published article and its supplementary information files or will be available from the authors upon reasonable request.

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

## Acknowledgements

We thank Dr Daniel Bernard for providing ID3 promoter-reporter construct and Dr Guohong Li for providing His-H2A.Z and H2B plasmids. We thank Dr Dawang Zhou for providing Vav-Cre transgenic mice. We thank Junfeng Hao, Yan Teng, Xudong Zhao, Xiaofei Guo and Junying Jia for technical assistance. We thank Xiang Shi and Liangming Yao for technical help and assistance with animal procedures. We thank Jing Li (Cnkingbio Company Ltd, Beijing, China) for technical support. This work was supported by the National Natural Science Foundation of China (31530093, 91640203, 81330047, 31429001, 31470864, 31670886) and the Strategic Priority Research Programs of the Chinese Academy of Sciences (XDB19030203).

## Author contributions

B.Y. and B.L. designed and performed experiments; B.Y. analysed data and wrote the paper; L.Y., G.H., L.H., X.Q. and P.Z. performed some experiments; J.W. crossed mice; P. X., S.W. and Y.D. analysed data. N.S. and J.Z. prepared mice and analysed data; Z.F. initiated the study, organized, designed and wrote the paper.

## Additional information

**Competing interests:** The authors declare no competing financial interests.

