## [Peer Review File · Nature Communications]

Reviewers' comments:

Reviewer #1 (SRCAP, chromatin remodeling)(Remarks to the Author):

This manuscript is largely beyond my areas of expertise. The only aspect that I can possibly comment on is with respect to the experiments directly revolving around SRCAP, which is essentially described in Figure 4. Overall, these experiments are properly executed and described, but the one issue is the very limited analysis of Pcid2 by yeast two hybrid. This is not a modern technology and there are likely many more Pcid2 interactors that the authors did not consider as a result of using this approach. Using an affinity purification and proteomics approach would be much more insightful and also looking at online databases where Pcid2 may have been analyzed previously. What the authors likely don't know is what are the most significant interactors of Pcid2 that may be more important for its role in lineage differentiation? Beyond this, I am not qualified to make any additional comments on the manuscript as the vast majority of the biology is well outside of my area of expertise.

Reviewer #2 (haematopoietic stem cell, epigenetic)(Remarks to the Author):

The paper by Ye et al. presents data showing that the proteins Pcid2 and ZNHIT1 interact to block the enzymatic activity of the "SNF2-related CBP activator protein" (SRCAP)-contained remodeling complex. The authors claim that this causes a block of the histone H2A/H2A.Z exchange at the nucleosomes of specific genes involved in lymphoid lineage commitment or determination.

Major points:

Fig. 1:

B: The authors use Pcid2 flox/flox;MxCre+ mice to conditionally delete Pcid2 in hematopoietic cells. Given the limitations and known artifacts that the Mx-Cre/pIpC system can cause in hematopoietic stem cells, it would be reassuring if other Cre strains were used to confirm the phenotype associated with the ablation of Pcid2 (for instance Vav-Cre or a Rosa-CreER with tamoxifen). If such data are available, they should be included.

D: Unclear from the Figure legend what is the %age shown is related to. Please clarify.

E: The authors make a claim about the expansion of thymocytes, but show very little information about this. The data in the suppl. File indicate an expansion of the DN population and it would be informative to look that the DN subsets to determine whether the source of the thymocytes expansion lies in altered proliferation and or survival of the cells making up the DN1-4 compartments, since the thymus is principally dependent on commitment and expansion of these cells; i.e. the D1/ETPs that migrate into the thymus from the bone marrow and the DN2 that is still cytokine dependent and the pre TCR expressing DN3 that are fully committed to the T-lineage.

G: The LMPP subset (Lin- Sca1+ c-Kit+ Flt3high) has to be included here since these cells are among the ones that still bear the potential for both myeloid and lymphoid differentiation and are a step further ahead compared to MPPs which are lower in Flt3. Similarly, the CLP subset is defined here by the authors without including Flt3 and other markers and should be gated at least with Flt3 marker (Lin- IL7R+ Flt3+Sca1med c-Kitmed).

H: Pro T and pro B cell subtypes are not defined well enough; the B220/CD43 gating itself is not sufficient. A fractionation per FACS according to the HARDY subsets is recommended here (e.g.: Hardy fractions A (pre-pro B cells, B220+ CD43+ BP-1- HSA-) and B (pro B cells, B220+ CD43+ BP-1- HSA+)) to exactly determine where exactly the Pcid2 deletion affects early B-cell development. Similarly: early T cell development has to be looked at more precisely and for this the DN subpopulation has to be analyzed (see above). The DN1 and DN2 subsets are still open to form other lineages and only DN3 are fully committed to T- lymphoid cells. The DN3 formation is a critical point in the generation of T- lymphoid cells, which is not addressed here. This can be done by FACS analysis.

Fig. 2:

C, D: Preferably LMPPs instead of MPPs (or both) should have been used in the OP9 DL1 culture system. The OP9 DL1 experiment shows the formation of the DN1-DN4 pre T cell subset. The total DN (CD44-CD25-) subset is increased in Pcid2 KO mice. This is of interest but is not followed further e.g. which subset DN1-DN4 is affected and whether the ELPs from the bone marrow that for the ETP/DN1 subset are reduced or increased. This would allow to determine where exactly the effect of Pcid2 ablation takes effect. In addition a culture with OP cells without Notch ligand has to be used under lymphoid and myeloid culture conditions to determine the potential of the MPPs to form B-cells and myeloid cells. These experiments taken together would provide a more complete picture of the lineage potential of the Pcid2 KO MPPs.

Fig. 3:

B: same comments as above for including the LMPP subset and the inclusion of missing markers into the gating of CLPs and MPPs.

C: the Gr-1/CD11b staining looks unusual. This staining allows to separate monocytes (CD11b+/Gr1lo) from granulocytes (CD11b+/Gr1hi) and would show whether the monocyte population and with this the granulocyte precursors are affected by Pcid2 ablation. Technical problem? Or use a different antibody combination.

Fig. 4:

D: Does Pcid2 interact with ZNHIT1 at endogenous expression levels in HSCs?

Fig. 5:

One would have expected to see a RNA seq profile and GSEA from MPPs (or better LMPPs) from wt and Pcid2 deficient mice and the enrichment of gene ontology pathways typical for lymphoid lineage cells and the loss of a myeloid signature. This would give a more complete information about the lineage decision process that is apparently disturbed in Pcid2 KO progenitors.

D: same comment as for Figure 2C: The total DN (CD44-CD25-) subset is measured here in different strains. While this is of interest, it is not followed up, which subset i.e. DN1-DN4 is affected and whether the ELPs from the bone marrow that for the ETP/DN1 subset are reduced or increased. This would allow to determine again exactly where the effect of Pcid2 ablation takes effect and would also allow to consolidate the finding of the interdependence of Pcid2 with Spi1.

G: The authors state that Pcid2 deficiency augmented chromatin accessibility to DNase I digestion at the promoters of these lymphoid determination genes such as Il7r and Ikzf1 (encoding Ikaros) (Fig. 5G). However, it seems that DNase I accessibility is in fact lower in Pcid2 deficient cells? Please clarify.

H: The luciferase result is intriguing and the question that has to be clarified is whether PU.1, SCRAP complex and Pcid2 and Znhit1 co-occupy the same sites or different sites of the IL-7R promoter-enhancer loci. It would have been useful to include a myeloid differentiation gene and show that accessibility is different than at lymphoid differentiation genes.

For Figure 5H: at least one myeloid specific promoter should be included in this type of analysis (e.g. MCSF-R) to show that lymphoid and myeloid determination genes are differentially regulated by PU.1, Pcid2 and Znhit1. This would allow closing in to propose a more precise model how Pcid2 fits into the already characterized regulatory network of transcription factors that regulate lineage decision of bi-potential myeloid/lymphoid progenitors.

J, K: same critique as for Fig. 2C. The OP9 and OP9-DL1 (or better OP9-DL4) system has to be used together and myeloid and lymphoid conditions to assess lineage potential of MPPs (or HSCs). As described here the B-lymphoid and myeloid compartment are not analyzed and are missing from the picture.

Fig. 6:

A: very little information is provided with regard to the generation of Znhit null cells, such as GFP FACS to follow the infection efficiency (e.g. as a suppl. file).

C, D: same critique as for Figure 2 C and Figure 5 J, K with regard to the analysis of the DN thymocytes subset.

Discussion:

As the authors discuss, one model of cell fate decision that is widely accepted and discussed is the antagonistic relationship between the PU.1 and GATA-1. In addition, PU.1 acts also by its expression level, i.e. high levels promote myeloid development and low levels B-cell development, but this is not addressed experimentally in a comprehensive manner.

A number of other factors are involved in this type of lineage decisions such as the factors Gfi1, Egr1 Nab2 and others. All these factors and the specific role of PU.1 are not addressed here, but are making up the present understanding how myeloid/lymphoid lineage decision is achieved. They could be discussed in more detail.

The authors also claim that Pcid2 acts as an upstream regulator of PU.1 to suppress its activity in MPPs, leading to restrict lymphoid lineage specification. Data in Fig. 5 support this, but it remains unclear how this is achieved exactly.

Reviewer #3 (cell fate, leukemia)(Remarks to the Author):

In their new study Drs. Ye, Fan and colleagues investigated the consequences of PCI domain-containing protein 2 (Pcid2) ablation on lineage determination of immature hematopoietic progenitor cells, and identify a novel molecular mechanism mediated by Pcid2-mediated epigenetic repression of lymphoid-lineage instructing gene loci that help governing lineage specification at the multipotent progenitor cell level.

Pcid2 has been shown to participate in the export of mRNA from the nucleus and preventing mRNA-mediated genomic instability. Moreover, loss of function studies demonstrated the role of this protein in promoting B cell survival via the regulation of cell cycle checkpoint modulators. Previous work from the same group revealed that Pcid2 governs self-renewal of embryonic stem cells. The role and molecular function of Pcid2 in hematopoietic lineage commitment had not been investigated before.

This new study provides a novel molecular mechanism for early lineage specification during hematopoiesis.

The investigators conducted a well-designed series of well-controlled, complementary ex vivo and vivo differentiation assays which revealed that induced and hematopoiesis-restricted ablation of Pcid2 leads to an increase in lymphoid cell production with a concomitant decrease in myeloid lineage generation. The authors provide interesting data attributing this lymphoid lineage bias to enhanced, cell intrinsic lymphoid differentiation commitment at the multipotent progenitor cell (MPP) level. The authors also revealed a novel regulatory mechanism by which Pcid2 suppresses gene expression via the modulation of SRCAP chromatin remodeling complex recruitment and H2A.Z deposition at loci driving lymphoid lineage specification at the MPP level. Interaction of Pcid2 with ZNHIT1 was found to prevent its binding to SCRAP and the exchange of H2A/H2A.Z via the inhibition of the complex's ATPase activity.

While the conclusions made by the investigators are in general well-founded and supported by a good body of quality data, I have some comments and suggestions to further substantiate the author's claims – which are especially focused on clarifying the molecular mechanism underlying the Pcid2-mediated repression of lymphoid genes.

Major critique:

- 1) Figure 2: The presented Pcid2 lentivirus-mediated gene rescue data are elegant, yet missing the essential information on the level of ectopic Pcid2 re-expression. Please include mRNA and where possible, protein quantification; same for PU.1 expression restoration;

- 2) Could the bias towards the lymphoid lineage be contributed to an increase in lymphoid-biased MPP subpopulations? The MPP population comprises a quite heterogenous population of lineage-biased cells (Oguro et al., Cell Stem Cell 2013). Could the authors provide a more differential analysis of MPP populations using CD229 and CD244 along with the already used SLAM markers?

- 3) How does Pcid2 affect SRCAP and lymphoid gene expression in more mature lymphoid cells?

- 4) How the specificity of Pcid2 in repressing lymphoid genes is achieved has not become extensively clear.
 - a. The authors provide convincing data on the significant increase of H2A.Z, SCRAP and PU.1 recruitment and concomitant depletion of H2A in Pcid2 KO on a set of hand-selected loci encoding lymphoid lineage-relevant genes. How do loci encoding myeloid lineage effector genes, many of which are also regulated by PU.1, behave?

 - b. The authors show expression data for a set of hand-curated lymphoid and myeloid genes; the investigators state on p.9 that these myeloid genes are not expressed in MPPs – despite the fact that Fig. 5F suggests mRNA expression. In any case, it is important to elucidate how expression of genes (potentially PU.1 regulated) contributing to lymphoid vs. myeloid lineage differentiation respond when Pcid2 is absent.

 - c. It is unlikely that the increase of PU.1 expression is the sole explanation for the increased lymphoid gene expression in Pcid2-KO MPPs. How do the authors explain increased PU.1 recruitment to lymphoid gene-encoding loci in absence of Pcid2? Is there perhaps another PU.1-co-operating TF that enables the preferential binding of PU.1 to these loci (rather than myeloid gene encoding regions), which then allows for SRCAP recruitment, H2A.Z deposition and the upregulation of gene expression when Pcid2 is absent?

Minor critique:

- 1) Figure 1/S1:

- a. Given the previously reported role of Pcid2 in B cell survival it would be helpful to include data on the frequency of B cell progenitor and precursor cells in Pcid2 KO mice (MX-1-Cre and Vav-Cre- induced); S1D shows increased frequencies of Pro-B cells, but a statistical analysis is lacking;
- b. What is the expression of Pcid2 in hematopoietic stem cells?
- c. p.5 Please include data on the phenotype of Vav-Cre deleted Pcid2 KO mice in the supplement;

2) Figure 2,5,6:

- a. Can the authors further specify what “CD44-CD25-“ cells are?
- b. p.6 please change title of paragraph stating the Pcid2-deficient HSCs, as the data provided focuses on MPPs.

3) FACS plots for myeloid progenitors (Lin-cKit+Sca-1-CD34 CD16/32; Fig. S1, 3) do not look normal – why is there no separation of CD34^{lo} and CD34^{high} cells; many events are beyond the y-axis and the gates placed for CMPs and GMPs do not agree with previous reports. Please show a more reliable data for the analysis of these progenitors and eventually revise the quantitative analysis of these progenitors in Pcid2 WT and KO animals;

4) Fig.3:

- a. X-axis label: I assume you mean CD45.1 (recipients are CD45.1/45.2 double positive, but this should not be stated as the parameter measured on the x-axis). Please correct, this is highly confusing otherwise.

5) Data sets showing non-significant differences between Pcid2 WT and KO should be moved to the supplement;

6) Fig. 4C. Can the authors include a z-stack projection to unequivocally demonstrate nuclear and co-localization of Pcid2 and ZNHIT1?

7) ChIP experiments need appropriate negative controls (e.g. IgG, non-occupied loci etc.) and a better explanation of how the “fold enrichment” values were normalized.

Reviewer #4 (NcoR1, nuclear receptor)(Remarks to the Author):

ATP-dependent chromatin remodeling machines including SWI/SNF and INO80 are evolutionarily conserved from yeast to human and involved in cell fate determination. Especially, previous work from this group has implicated NURF chromatin remodeling complex as an important factor for stem cell self-renewal. This article, Ye et al. also describe chromatin remodeling complex dependent cell fate decision in immune cell system.

In this paper, the authors suggest an interesting mechanism of lymphoid lineage commitment. As a follow-up study of previous finding, role of Pcid as a self-renewal factor in ES cells. The authors further investigated the Pcid function in other cells types. They found that Pcid is important protein for myeloid vs lymphoid decision in multipotent progenitor cells. Interestingly, Pcid can bind to ZNHIT1 and inhibit ZNHIT1 from making complex with SRCAP. Therefore, in the absence of Pcid, SRCAP excessively interacts with ZNHIT1 and exchanges H2A to H2AZ at the promoter of lymphoid-determining genes. Consequently, the loci open more, got more transcription factor PU.1 binding and this leads high expression of lymphoid determination genes. KO of ZNHIT1 decrease H2A exchange to H2A.z and lymphoid lineage commitment. The opposite phenotype between Pcid KO and ZNHIT1 is well matched to the authors' model that Pcid2 binds to ZNHIT1 to impair the SRCAP complex activity that suppresses the expression of lymphoid determination genes.

All the data is very clean and well support the authors point of view. The authors used a well designed conditional KO mouse system, bone marrow transfer, in vitro differentiation, CHIP and other epigenetic methods. If the questions listed below are answered I am pleased to recommend acceptance of this article.

Major issue

1. The authors' BM transfer approach should preclude any controversy. However, I am still concern about the effect of ICI. Still the authors are using the immune system, one cannot exclude the possible unintended effect of IFN. Authors should display other KO system (VAV-cre).
2. Although the authors well performed CHIP-qPCR at several loci of lymphoid and myeloid determining genes and obtained consistent results, it would be far better to provide genome-wide data of SRCAP and chromosome openness. This will broaden our understanding why chromatin remodeling complex behaves as it has transcription factor specificity during immune cell lineage commitment.

With these questions address, this manuscript will provide a valuable contribution.

Point-by-point response to the reviewers' comments

Reviewer #1:

This manuscript is largely beyond my areas of expertise. The only aspect that I can possibly comment on is with respect to the experiments directly revolving around SRCAP, which is essentially described in Figure 4. Overall, these experiments are properly executed and described, but the one issue is the very limited analysis of Pcid2 by yeast two hybrid. This is not a modern technology and there are likely many more Pcid2 interactors that the authors did not consider as a result of using this approach. Using an affinity purification and proteomics approach would be much more insightful and also looking at online databases where Pcid2 may have been analyzed previously. What the authors likely don't know is what are the most significant interactors of Pcid2 that may be more important for its role in lineage differentiation? Beyond this, I am not qualified to make any additional comments on the manuscript as the vast majority of the biology is well outside of my area of expertise.

Answer: This is a good suggestion. Through yeast two-hybrid screening, 57 positive clones were screened out. Of these positive clones, 13 clones were identified to be ZNHIT1. More importantly, the association of Pcid2 with ZNHIT1 was further validated by co-IP assays (new Fig. 4D). We depleted ZNHIT1 and other known Pcid2 interactors (DSS1, EID1, GANP and BRCA2) in LSKs, followed by transplantation assays. We observed that engraftment with Znhit1-depleted MPPs abrogated lymphoid lineage commitment capacity (Attached Figure 1A). However, engraftment with knockdowns of other genes did not affect this lineage commitment (Attached Figure 1A). In this study, we thus focused on the interactor ZNHIT1 of Pcid2 for the regulation of lymphoid lineage specification. We also conducted affinity chromatography by using anti-Pcid2 antibody. We observed that Pcid2 could associate with ZNHIT1 in BM lysates (Attached Figure 1B).

Reviewer #2:

The paper by Ye et al. presents data showing that the proteins Pcid2 and ZNHIT1 interact to block the enzymatic activity of the "SNF2-related CBP activator protein" (SRCAP)-contained remodeling complex. The authors claim that this causes a block of the histone H2A/H2A.Z exchange at the nucleosomes of specific genes involved in lymphoid lineage commitment or determination.

Major points:

Fig. 1:

B: The authors use Pcid2 flox/flox;MxCre+ mice to conditionally delete Pcid2 in hematopoietic cells. Given the limitations and known artifacts that the Mx-Cre/plpC system can cause in hematopoietic stem cells, it would be reassuring if other Cre strains were used to confirm the phenotype associated with the ablation of Pcid2 (for instance Vav-Cre or a Rosa-CreER with tamoxifen). If such data are available, they should be included.

Answer: This is a good point. We also repeated these experiments by crossing Pcid2^{flox/flox} mice with Vav-Cre mice. We observed that Pcid2^{flox/flox};Vav-Cre⁺ mice

displayed the same phenotype as that of *Pcid2^{flox/flox};MxCre⁺* mice post poly (I:C) treatment. We provided these data in the new Fig. S1D and Fig. S2B.

D: Unclear from the Figure legend what is the %age shown is related to. Please clarify.

Answer: We clarified this issue in the respective figure legend.

E: The authors make a claim about the expansion of thymocytes, but show very little information about this. The data in the suppl. File indicate an expansion of the DN population and it would be informative to look that the DN subsets to determine whether the source of the thymocytes expansion lies in altered proliferation and or survival of the cells making up the DN1-4 compartments, since the thymus is principally dependent on commitment and expansion of these cells; i.e. the D1/ETPs that migrate into the thymus from the bone marrow and the DN2 that is still cytokine dependent and the pre TCR expressing DN3 that are fully committed to the T-lineage.

Answer: We analyzed DN1-4 compartment cells in *Pcid2^{+/+}* and *Pcid2^{-/-}* thymocytes. We noticed that numbers of DN1-4 compartment cells in *Pcid2^{-/-}* thymocytes were all increased compared to *Pcid2^{+/+}* thymocytes (new Fig. 1H), but displayed similar composition ratios to those of *Pcid2^{+/+}* thymocytes (new Fig. S2C). Moreover, DN1-4 compartment cells in *Pcid2^{+/+}* and *Pcid2^{-/-}* thymocytes showed similar proliferation and survival states as well (Attached Figure 2A, B). These data suggest that *Pcid2* knockout does not affect the proliferation and survival states of DN1-4 compartment cells in the thymus.

G: The LMPP subset (Lin- Sca1+ c-Kit+ Flt3high) has to be included here since these cells are among the ones that still bear the potential for both myeloid and lymphoid differentiation and are a step further ahead compared to MPPs which are lower in Flt3. Similarly, the CLP subset is defined here by the authors without including Flt3 and other markers and should be gated at least with Flt3 marker (Lin- IL7R+ Flt3+Sca1med c-Kitmed).

Answer: This is a good suggestion. As shown in the new Fig. 1G and Fig. S2A, we provided the LMPP population with Flt3 gating. We found that LMPP subset in *Pcid2^{-/-}* mice was not significantly changed compared to *Pcid2^{+/+}* mice.

H: Pro T and pro B cell subtypes are not defined well enough; the B220/CD43 gating itself is not sufficient. A fractionation per FACS according to the HARDY subsets is recommended here (e.g.: Hardy fractions A (pre-pro B cells, B220+ CD43+ BP-1- HSA-) and B (pro B cells, B220+ CD43+ BP-1- HSA+)) to exactly determine where exactly the *Pcid2* deletion affects early B-cell development. Similarly: early T cell development has to be looked at more precisely and for this the DN subpopulation has to be analyzed (see above). The DN1 and DN2 subsets are still open to form other lineages and only DN3 are fully committed to T- lymphoid cells. The DN3 formation is a critical point in the generation of T- lymphoid cells, which is not addressed here. This can be done by FACS analysis.

Answer: As shown in the new Fig. 1I and Fig. S2C, we analyzed the pre-pro B cell subtype by using B220+ CD43+ BP-1- HSA+ staining strategy. As addressed in the Fig. 1E, we analyzed DN1-4 compartment cells in *Pcid2*^{+/+} and *Pcid2*^{-/-} thymocytes. (new Fig. 1H and Fig. S2C).

Fig. 2:

C, D: Preferably LMPPs instead of MPPs (or both) should have been used in the OP9 DL1 culture system. The OP9 DL1 experiment shows the formation of the DN1-DN4 pre T cell subset. The total DN (CD44-CD25-) subset is increased in *Pcid2* KO mice. This is of interest but is not followed further e.g. which subset DN1-DN4 is affected and whether the ELPs from the bone marrow that for the ETP/DN1 subset are reduced or increased. This would allow to determine where exactly the effect of *Pcid2* ablation takes effect. In addition a culture with OP cells without Notch ligand has to be used under lymphoid and myeloid culture conditions to determine the potential of the MPPs to form B-cells and myeloid cells. These experiments taken together would provide a more complete picture of the lineage potential of the *Pcid2* KO MPPs.

Answer: This is a very good point. We repeated these experiments by incubation of LMPPs with OP9-DL1 cells. As expected, all DN1-4 compartment cells of *Pcid2*^{-/-} LMPPs were increased compared to *Pcid2*^{+/+} LMPPs (new Fig. 2C). However, the ratio of each DN subset between *Pcid2*^{-/-} and *Pcid2*^{+/+} LMPPs was not changed. These results suggest that *Pcid2* knockout does not affect the stages of DN1-4s. In addition, we incubated MPPs with OP9 feeder cells to form B-cells and myeloid cells. We found *Pcid2*^{-/-} MPPs generated reduced B cells and myeloid cells compared with *Pcid2*^{+/+} MPPs (new Fig. 2E), which were consistent with Nakaya's report (Nakaya T. et al. JI, 2010).

Fig. 3:

B: same comments as above for including the LMPP subset and the inclusion of missing markers into the gating of CLPs and MPPs.

Answer: We included these data in the new Fig. 3B.

C: the Gr-1/CD11b staining looks unusual. This staining allows to separate monocytes (CD11b+/Gr1lo) from granulocytes (CD11b+/Gr1hi) and would show whether the monocyte population and with this the granulocyte precursors are affected by *Pcid2* ablation. Technical problem? Or use a different antibody combination.

Answer: We repeated these experiments with new antibodies and provided better data as shown in the new Fig. 3C.

Fig. 4:

D: Does *Pcid2* interact with ZNHIT1 at endogenous expression levels in HSCs?

Answer: Since the number of HSCs is very low, we could not get enough cells (1X10⁶) for co-IP assays. We then used LSKs for co-IP assays. As shown in the new Fig. 4 (D), we found that anti-*Pcid2* antibody could precipitate ZNHIT1 from cell lysates of LSKs, suggesting that *Pcid2* interacts with ZNHIT1 at endogenous expression levels in LSKs.

Fig. 5:

One would have expected to see a RNA seq profile and GSEA from MPPs (or better LMPPs) from wt and *Pcid2* deficient mice and the enrichment of gene ontology pathways typical for lymphoid lineage cells and the loss of a myeloid signature. This would give a more complete information about the lineage decision process that is apparently disturbed in *Pcid2* KO progenitors.

Answer: This is a good suggestion. For RNA seq profile assays, at least 5×10^6 cells are needed for RNA extraction. Even in WT mice, the number of LMPPs is about 2×10^4 and MPPs around 5×10^4 . Since the numbers of LMPPs or MPPs were dramatically reduced in *Pcid2* deficient mice, we could not obtain enough cell numbers to perform RNA seq assays.

D: same comment as for Figure 2C: The total DN (CD44-CD25-) subset is measured here in different strains. While this is of interest, it is not followed up, which subset i.e. DN1-DN4 is affected and whether the ELPs from the bone marrow that for the ETP/DN1 subset are reduced or increased. This would allow to determine again exactly where the effect of *Pcid2* ablation takes effect and would also allow to consolidate the finding of the interdependence of *Pcid2* with *Spi1*.

Answer: Fig. 5D shows co-IP data. This issue should point to the Fig. 5 J. We provided these respective data in the new Fig. 5J.

G: The authors state that *Pcid2* deficiency augmented chromatin accessibility to DNase I digestion at the promoters of these lymphoid determination genes such as *Ii7r* and *Ikzf1* (encoding Ikaros) (Fig. 5G). However, it seems that DNase I accessibility is in fact lower in *Pcid2* deficient cells? Please clarify.

Answer: We are sorry to make wrong labels in this figure. We changed these labels accordingly.

H: The luciferase result is intriguing and the question that has to be clarified is whether PU.1, SRCAP complex and *Pcid2* and *Znhit1* co-occupy the same sites or different sites of the IL-7R promoter-enhancer loci. It would have been useful to include a myeloid differentiation gene and show that accessibility is different than at lymphoid differentiation genes.

Answer: This is a good suggestion. We performed fragment mapping on the IL-7R promoter-enhancer loci. We observed that PU.1, SRCAP and ZNHIT1 were co-occupied at the same site of IL-7R promoter-enhancer locus (-1400~-1200), whereas *Pcid2* did not bind to the same locus (Fig. S4B). The binding of ZNHIT1 to SRCAP prevented it from binding to *Pcid2*. Meanwhile, we used the myeloid differentiation gene *Csf1r* as an assay control. *Csf1r* harbors a PU.1 binding site in its promoter region. We noticed that PU.1 was accumulated at the position of -200~0 on the *Csf1r* promoter (Fig. S4B). However, SRCAP and ZNHIT1 were not co-occupied at the same locus on *Csf1r* promoter-enhancer region (Fig. S4B). We addressed this issue in the revised text.

For Figure 5H: at least one myeloid specific promoter should be included in this type of analysis (e.g. MCSF-R) to show that lymphoid and myeloid determination genes are differentially regulated by PU.1, Pcid2 and Znhit1. This would allow closing in to propose a more precise model how Pcid2 fits into the already characterized regulatory network of transcription factors that regulate lineage decision of bi-potential myeloid/lymphoid progenitors.

Answer: We analyzed *Csf1r* promoter transcription activity by luciferase assay. We noticed that depletion of *Pcid2*, *ZNHIT1* or *SRCAP* could not suppress *Csf1r* transcription (new Fig. S4C). These data suggest that lymphoid and myeloid determination genes are differentially regulated by PU.1, *Pcid2* and *Znhit1*. We stated this point in the revised manuscript.

J, K: same critique as for Fig. 2C. The OP9 and OP9-DL1 (or better OP9-DL4) system has to be used together and myeloid and lymphoid conditions to assess lineage potential of MPPs (or HSCs). As described here the B-lymphoid and myeloid compartment are not analyzed and are missing from the picture.

Answer: We provided these data in the new Fig. 5L.

Fig. 6.

A: very little information is provided with regard to the generation of *Znhit* null cells, such as GFP FACS to follow the infection efficiency (e.g. as a suppl. file).

Answer: We provided the infection efficiency in the new Fig. 6A.

C, D: same critique as for Figure 2 C and Figure 5J, K with regard to the analysis of the DN thymocytes subset.

Answer: We provided these data in the new Fig. 6C.

Discussion:

As the authors discuss, one model of cell fate decision that is widely accepted and discussed is the antagonistic relationship between the PU.1 and GATA-1. In addition, PU.1 acts also by its expression level, i.e. high levels promote myeloid development and low levels B-cell development, but this is not addressed experimentally in a comprehensive manner.

Answer: We discussed this point in the discussion section.

A number of other factors are involved in this type of lineage decisions such as the factors *Gfi1*, *Egr1*, *Nab2* and others. All these factors and the specific role of PU.1 are not addressed here, but are making up the present understanding how myeloid/lymphoid lineage decision is achieved. They could be discussed in more detail.

Answer: We discussed this issue in the discussion section.

The authors also claim that *Pcid2* acts as an upstream regulator of PU.1 to suppress its activity in MPPs, leading to restrict lymphoid lineage specification. Data in Fig. 5 support this, but it remains unclear how this is achieved exactly.

Answer: We discussed this point in the discussion section.

Reviewer #3:

Major critique:

1) Figure 2: The presented *Pcid2* lentivirus-mediated gene rescue data are elegant, yet missing the essential information on the level of ectopic *Pcid2* re-expression. Please include mRNA and where possible, protein quantification; same for PU.1 expression restoration;

Answer: We provided these results in the new Fig. 2G and Fig. S4D.

2) Could the bias towards the lymphoid lineage be contributed to an increase in lymphoid-biased MPP subpopulations? The MPP population comprises a quite heterogeneous population of lineage-biased cells (Oguro et al., Cell Stem Cell 2013). Could the authors provide a more differential analysis of MPP populations using CD229 and CD244 along with the already used SLAM markers?

Answer: This is a good point. We analyzed MPP populations using CD229, CD244 along with SLAM markers. Based on the report of Oguro et al, MPP-1 is classified by CD150⁺CD48⁻CD229⁻CD244⁻ LSK, MPP-2 with CD150⁺CD48⁻CD229⁺CD244⁻ LSK, MPP-3 with CD150⁺CD48⁻CD229⁺CD244⁺ LSK. We found that numbers of MPP-1, MPP-2, and MPP-3 were all reduced in *Pcid2* KO mice (new Fig. S2B). We are still exploring which MPP subpopulation contributes to the biased lymphoid lineage commitment by *Pcid2*.

3) How does *Pcid2* affect SRCAP and lymphoid gene expression in more mature lymphoid cells?

Answer: We found that *Pcid2* was lowly expressed in mature lymphocytes (new Fig. S1A). *Pcid2* knockout did not affect SRCAP ATPase activity (Attached Fig. 3A) and lymphoid gene expression (Attached Fig. 3B) in CD3⁺ T cells. It still needs to be further investigated about whether *Pcid2* affects the functions of mature T cells.

4) How the specificity of *Pcid2* in repressing lymphoid genes is achieved has not become extensively clear.

a. The authors provide convincing data on the significant increase of H2A.Z, SRCAP and PU.1 recruitment and concomitant depletion of H2A in *Pcid2* KO on a set of hand-selected loci encoding lymphoid lineage-relevant genes. How do loci encoding myeloid lineage effector genes, many of which are also regulated by PU.1, behave?

Answer: We sorted MPPs from *Pcid2*^{-/-} and *Pcid2*^{+/+} mice to conduct CHIP assays. We observed that *Pcid2* deletion did not cause H2A.Z, SRCAP and PU.1 deposition onto the promoters of myeloid lineage effector genes (Fig. S4A). These data suggest that PU.1 behaves differently in the loci of lymphoid and myeloid lineage effector genes, which may be due to their unique locus tertiary structures.

b. The authors show expression data for a set of hand-curated lymphoid and myeloid genes; the investigators state on p.9 that these myeloid genes are not expressed in MPPs – despite the fact that Fig. 5F suggests mRNA expression. In any case, it is

important to elucidate how expression of genes (potentially PU.1 regulated) contributing to lymphoid vs. myeloid lineage differentiation respond when *Pcid2* is absent.

Answer: We are sorry that we made wrong descriptions in Fig. 5F. The lymphoid determination genes were upregulated in *Pcid2*^{-/-} MPPs, whereas the expression levels of myeloid determination gene was not changed in *Pcid2*^{-/-} MPPs compared with *Pcid2*^{+/+} MPPs. We revised this sentence accordingly.

c. It is unlikely that the increase of PU.1 expression is the sole explanation for the increased lymphoid gene expression in *Pcid2*-KO MPPs. How do the authors explain increased PU.1 recruitment to lymphoid gene-encoding loci in absence of *Pcid2*? Is there perhaps another PU.1-co-operating TF that enables the preferential binding of PU.1 to these loci (rather than myeloid gene encoding regions), which then allows for SRCAP recruitment, H2A.Z deposition and the upregulation of gene expression when *Pcid2* is absent?

Answer: Based on our experimental data, *Pcid2* interacts with ZNHIT1 to block its association with SRCAP for shutdown of PU.1 expression in WT MPPs. In *Pcid2* KO MPPs, released ZNHIT1 associates with SRCAP to activate PU.1 expression, leading to lymphoid gene transcription. It is possible that another PU.1-co-operating TF facilitates the preferential binding of PU.1 to lymphoid gene promoter loci. We addressed this issue in the discussion section.

Minor critique:

1) Figure 1/S1:

a. Given the previously reported role of *Pcid2* in B cell survival it would be helpful to include data on the frequency of B cell progenitor and precursor cells in *Pcid2* KO mice (MX-1-Cre and Vav-Cre- induced); S1D shows increased frequencies of Pro-B cells, but a statistical analysis is lacking;

Answer: We added these data in the new Fig. 1I and Fig. S2C.

b. What is the expression of *Pcid2* in hematopoietic stem cells?

Answer: *Pcid2* is also highly expressed in HSCs (new Fig. S1A and new Fig. 4C).

c. p.5 Please include data on the phenotype of Vav-Cre deleted *Pcid2* KO mice in the supplement;

Answer: We provided these data in the new Fig. S1D

2) Figure 2,5,6:

a. Can the authors further specify what "CD44⁺CD25⁻" cells are?

Answer: In early T cell development, committed lymphoid progenitors migrate from bone marrow to the thymus. Early committed T cells do not express TCR, CD4 and CD8 surface markers that termed as double-negative (DN) thymocytes. DN cells can be further divided into four subset stages (DN1, CD44⁺CD25⁻; DN2, CD44⁺CD25⁺; DN3, CD44⁺CD25⁺; and DN4, CD44⁺CD25⁻). DN4 cells with pre-TCR expression can

differentiate into double positive (DP, CD4⁺CD8⁺) T cells for consequent mature T cell development.

b. p.6 please change title of paragraph stating the *Pcid2*-deficient HSCs, as the data provided focuses on MPPs.

Answer: We corrected this wording in the text.

3) FACS plots for myeloid progenitors (Lin-cKit+Sca-1-CD34^{lo}CD16/32; Fig. S1, 3) do not look normal – why is there no separation of CD34^{lo} and CD34^{high} cells; many events are beyond the y-axis and the gates placed for CMPs and GMPs do not agree with previous reports. Please show a more reliable data for the analysis of these progenitors and eventually revise the quantitative analysis of these progenitors in *Pcid2* WT and KO animals;

Answer: We repeated these experiments using new anti-CD34 antibody and provided better data in the according figures (new Fig. 3B and S2D).

4) Fig.3:

a. X-axis label: I assume you mean CD45.1 (recipients are CD45.1/45.2 double positive, but this should not be stated as the parameter measured on the x-axis). Please correct, this is highly confusing otherwise.

Answer: We changed it accordingly.

5) Data sets showing non-significant differences between *Pcid2* WT and KO should be moved to the supplement;

Answer: We moved these data to the supplementary figures (new Fig. S3A-C).

6) Fig. 4C. Can the authors include a z-stack projection to unequivocally demonstrate nuclear and co-localization of *Pcid2* and ZNHIT1?

Answer: We provided co-localization data using z-stack projection in the new Fig. 4C.

7) ChIP experiments need appropriate negative controls (e.g. IgG, non-occupied loci etc.) and a better explanation of how the “fold enrichment” values were normalized.

Answer: We provided negative controls in the respective figures (new Fig. 5A-C, 5E, 6B, S4A-B) and stated calculation methods of “fold enrichment” in the corresponding legends.

Reviewer #4:

Major issue:

1. The authors' BM transfer approach should preclude any controversy. However, I am still concern about the effect of ICI. Still the authors are using the immune system, one cannot exclude the possible unintended effect of IFN. Authors should display other KO system (VAV-cre).

Answer: This is good suggestion. We also repeated these experiments by crossing *Pcid2*^{flox/flox} mice with *Vav-Cre* mice. We observed that *Pcid2*^{flox/flox}; *Vav-Cre*⁺

mice displayed the same phenotype as that of *Pcid2*^{flox/flox}; *MxCre*⁺ mice post poly (I:C) treatment. We provided these data in the new Fig. S1D and Fig. S2B.

2. Although the authors well performed CHIP-qPCR at several loci of lymphoid and myeloid determining genes and obtained consistent results, it would be far better to provide genome-wide data of SRCAP and chromosome openness. This will broaden our understanding why chromatin remodeling complex behaves as it has transcription factor specificity during immune cell lineage commitment.

Answer: This is a very good point. For genome-wide chromosome openness assays, at least 5×10^6 cells are needed for DNA extraction. Even in WT mice, the number of MPPs is around 5×10^4 . Since the number of MPPs was dramatically reduced in *Pcid2* deficient mice, we could not obtain enough cell numbers to perform genome-wide chromosome assays at this stage.

Attached figure 1. ZNHIT1 is identified as a significant interactor of Pcid2 in lineage differentiation. (A) The shRNA oligoes encoding target sequences against mDss1, mEid1, mGanp and mBrca2 were cloned into MSCV-LTRmiR30-PIG vector and prepared for retrovirus. Mouse BM LSKs were sorted, infected by shRNA-retrovirus and transplanted to lethally irradiated recipient mice. BM cells of recipients were analyzed 4 weeks post transplantation. (B) Mouse bone marrow cells were lysed and immunoprecipitated with anti-Pcid2 antibody followed with mass spectrometry identification.

Attached figure 2. *Pcid2* deficiency does not affect the process of T cell development. (A) DN cell expansion was further analyzed with CD44/CD25 antibody staining followed with flow cytometry. The frequency of each DN subset between *Pcid2*^{-/-} and *Pcid2*^{+/+} sample was not significantly changed. (B) Cell cycle analysis of MPPs from *Pcid2*^{+/+} and *Pcid2*^{-/-} mice. Cells were stained by Ki67 followed by flow cytometry. (C) DN1-DN4 cells were sorted and apoptosis was detected by immunoblot using antibody against active -caspase 3.

Attached figure 3. *Pcid2* deficiency did not affect SRCAP ATPase activity and lymphoid gene expression in mature lymphoid cell. (A) *Pcid2*^{+/+} and *Pcid2*^{-/-} CD3⁺ T cell were sorted from thymus. T cell lysates were immunoprecipitated by anti-SRCAP antibody, followed by detection of ATPase activities. (B) CD3⁺ T cell were sorted by FACS. Total RNA was extracted from indicated cells and analyzed by real time-qPCR.

Reviewers' comments:

Reviewer #2 (Remarks to the Author):

The authors have done a substantial amount of new experimentation and have addressed almost all of the concerns raised. A few minor points remain:

The DN1-4 profile in Fig S2C looks very unusual. The % DN1 shown here cannot be a wt value. The authors should verify if the wt control is indeed the correct mouse.

For instance the values in Fig 6C do not seem to match the percentages in this Figure for the DN1-4 profiles.

The new OP9 experiments clearly suggests that B-cells and myeloid cells are affected but not so much the T-cell compartment. Maybe this could be stated more clearly.

The Mac-1/Gr1 staining (Fig. 3C) still looks unusual – the authors should check this staining again and compare to a published wt image from the literature for orientation.

Reviewer #3 (Remarks to the Author):

The authors have adequately addressed my points of critique and added important new data substantiating their central claims in the revised manuscript. I have no further comments, or suggestions.

Reviewer #4 (Remarks to the Author):

The revised manuscript by Ye et al is greatly improved beyond the initial submission. The more extensive number of crosses has made the basic alterations in myeloid vs lymphoid lineage

specification more rigorous. . While it is rather unfortunate that expression studies are not extended by global genomic approaches, the selected gene targets evaluated do make a strong case. The biochemical aspects of the study are suggestive, but do fall short on the type of detail one might prefer for the case being made that the major effect of Pcid2 is based on its interaction with ZNHIT1, identified initially in the 2 hybrid screen and then by initial interaction assays. The model that the consequence of this interaction is to block the ATPase activity of the SRCAP complex is supported by their biochemical mixing experiments, but even though they do provide some data that this affects H2A/H2AZ exchange at nucleosomes of several lymphoid determination genes, the conclusion that this represents a clear mechanism for the effects of lineage maturation is rather premature. Especially as PU.1 functions primarily at the level of gene enhancers, and they exclusively investigate picked promoters, the idea that this explains the similar phenotypes of PU.1 and Pcid2 deletion seems too bold in light of their data. I admire the depth of their mouse cross data and the initial data about Pcid2/ZNHIT1 interactions, and the challenging nucleosome exchange assays they perform, I think that it is important to modify the conclusions as expressed in the Abstract to fit the actual available data. I think, with a toning down of these conclusions that the manuscript is suitable for presentation in Nature Communications at this point, hoping that global genomic approaches will clarify some of the models raised in the future.

Point-by-point response to the reviewers' comments

Reviewer #2

The authors have done a substantial amount of new experimentation and have addressed almost all of the concerns raised. A few minor points remain:

The DN1-4 profile in Fig S2C looks very unusual. The % DN1 shown here cannot be a wt value. The authors should verify if the wt control is indeed the correct mouse. For instance the values in Fig 6C do not seem to match the percentages in this Figure for the DN1-4 profiles.

Answer: We repeated this experiment with new antibody combinations and provided better data in the new Fig. S2C.

The new OP9 experiments clearly suggest that B-cells and myeloid cells are affected but not so much the T-cell compartment. Maybe this could be stated more clearly.

Answer: We addressed this point in the revised text.

The Mac-1/Gr1 staining (Fig. 3C) still looks unusual – the authors should check this staining again and compare to a published wt image from the literature for orientation.

Answer: We repeated this experiment using new anti-Gr1 antibody and provided better data in the new Fig. 3C.

Reviewer #4

The revised manuscript by Ye et al is greatly improved beyond the initial submission. The more extensive number of crosses has made the basic alterations in myeloid vs lymphoid lineage specification more rigorous. While it is rather unfortunate that expression studies are not extended by global genomic approaches, the selected gene targets evaluated do make a strong case. The biochemical aspects of the study are suggestive, but do fall short on the type of detail one might prefer for the case being made that the major effect of Pcid2 is based on its interaction with ZNHIT1, identified initially in the 2 hybrid screen and then by initial interaction assays. The model that the consequence of this interaction is to block the ATPase activity of the SRCAP complex is supported by their biochemical mixing experiments, but even though they do provide some data that this affects H2A/H2AZ exchange at nucleosomes of several lymphoid determination genes, the conclusion that this represents a clear mechanism for the effects of lineage maturation is rather premature. Especially as PU.1 functions primarily at the level of gene enhancers, and they exclusively investigate picked promoters, the idea that this explains the similar phenotypes of PU.1 and Pcid2 deletion seems too bold in light of their data. I admire the depth of their mouse cross data and the initial data about Pcid2/ZNHIT1 interactions, and the challenging nucleosome exchange assays they perform, I think that it is important to modify the conclusions as expressed in the Abstract to fit the actual available data. I think, with a toning down of these conclusions that the manuscript is suitable for presentation in Nature Communications at this point, hoping that global genomic approaches will clarify some of the models raised in the future.

Answer: We appreciate the comment of this reviewer. We revised our description of conclusions in the abstract and respective text as suggested.

REVIEWERS' COMMENTS:

Reviewer #4 (Remarks to the Author):

The authors have well addressed all critique and the concerns raised.

Point-by-point response to the reviewers' comments

Reviewer #4

The authors have well addressed all critique and the concerns raised.

Answer: We thank the reviewer's comments.